# Analysis of the PcrA-RNA polymerase complex reveals a helicase interaction motif and a role for PcrA/UvrD helicase in the suppression of R-loops

Inigo Urrutia-Irazabal[1], James R Ault[2], Frank Sobott[2], Nigel J Savery[1], Mark S Dillingham[1]*

[1]DNA:Protein Interactions Unit, School of Biochemistry, University of Bristol. Biomedical Sciences Building, University Walk, Bristol, United Kingdom; [2]Astbury Centre for Structural Molecular Biology, School of Molecular and Cellular Biology, University of Leeds, Leeds, United Kingdom

**Abstract** The PcrA/UvrD helicase binds directly to RNA polymerase (RNAP) but the structural basis for this interaction and its functional significance have remained unclear. In this work, we used biochemical assays and hydrogen-deuterium exchange coupled to mass spectrometry to study the PcrA-RNAP complex. We find that PcrA binds tightly to a transcription elongation complex in a manner dependent on protein:protein interaction with the conserved PcrA C-terminal Tudor domain. The helicase binds predominantly to two positions on the surface of RNAP. The PcrA C-terminal domain engages a conserved region in a lineage-specific insert within the β subunit which we identify as a helicase interaction motif present in many other PcrA partner proteins, including the nucleotide excision repair factor UvrB. The catalytic core of the helicase binds near the RNA and DNA exit channels and blocking PcrA activity in vivo leads to the accumulation of R-loops. We propose a role for PcrA as an R-loop suppression factor that helps to minimize conflicts between transcription and other processes on DNA including replication.

\*For correspondence:
mark.dillingham@bristol.ac.uk

**Competing interests:** The authors declare that no competing interests exist.

## Introduction

Helicases are conserved proteins found in all kingdoms of life. They are involved in a wide variety of DNA transactions including replication, repair, recombination, and transcription. Structural and bio-informatic analyses have classified helicases into several superfamilies (SF) (*Gorbalenya and Koonin, 1993*; *Singleton et al., 2007*). Amongst the best-studied systems, particularly from a mechanistic point of view, are the UvrD-like family of SF1 enzymes, which include UvrD and its close cousin Rep from *E. coli*, as well as PcrA (a UvrD orthologue) from *B. subtilis* and related organisms. Structural and biochemical analyses of the helicases in this family revealed conserved mechanisms for ATP hydrolysis, DNA translocation, and DNA unwinding among the different members (*Lee and Yang, 2006*; *Velankar et al., 1999*).

The PcrA/UvrD helicase has roles in rolling circle replication, nucleotide excision repair (NER), mismatch repair (MMR), and homologous recombination (HR) (*Bruand and Ehrlich, 2000*; *Husain et al., 1985*; *Lahue et al., 1989*; *Merrikh et al., 2015*; *Petit et al., 1998*; *Zieg et al., 1978*). This adaptability is provided by the many different interaction partners, for example RepC/D, UvrB, MutL and RecA, which recruit the helicase to their respective pathways via physical interactions (*Hall et al., 1998*; *Machón et al., 2010*; *Manelyte et al., 2009*; *Veaute et al., 2005*). However, the structural basis for these interactions with different partner proteins remains largely unclear. Most recently, an interaction between PcrA/UvrD and RNAP has been identified, implying a role for PcrA in

transcription, but its function is not understood (*Delumeau et al., 2011*; *Epshtein et al., 2014*; *Gwynn et al., 2013*). Studies in *E. coli* showed that UvrD can act as an accessory helicase which helps to minimise conflicts between the replisome and replication fork barriers including transcription complexes (*Boubakri et al., 2010*; *Guy et al., 2009*; *Hawkins et al., 2019*). The interaction was also reported to facilitate an alternative pathway for transcription-coupled nucleotide excision repair (TC-NER) in which UvrD removes RNAP from lesions by promoting backtracking and then helps to recruit NER factors to repair the damage (*Epshtein et al., 2014*; *Kamarthapu and Nudler, 2015*). However, the existence of this new TC-NER pathway (separate from that promoted by the canonical transcription-repair coupling factor Mfd) is debated, because high-throughput sequencing of the oligonucleotides excised during NER showed no evidence for strand-specific repair promoted by UvrD (*Adebali et al., 2017a*). In the Gram-(+)ve model organism *Bacillus subtilis*, PcrA was found to be enriched at highly transcribed regions of the genome and, in common with its orthologue UvrD, shown to alleviate replication-transcription conflicts (*Merrikh et al., 2015*). Furthermore, genetic analyses of PcrA place this helicase at the interface of replication, repair, transcription, and chromosome segregation (*Moreno-Del Alamo et al., 2020*; *Petit et al., 1998*). However, the mechanistic details of how PcrA/UvrD might help to promote the replication and/or repair of DNA on actively transcribed DNA remain unclear.

One nucleic acid intermediate that impacts on each of these key DNA transactions is the R-loop; a three-stranded structure in which a single RNA strand hybridises to a DNA duplex displacing an equivalent region of ssDNA (*Aguilera and García-Muse, 2012*; *Crossley et al., 2019*). R-loops can be formed when an RNA transcript re-hybridises to template DNA behind RNA polymerase. Once formed, R-loops are relatively stable structures which require active removal by enzymes such as RNases to prevent interference with other processes on DNA and preserve genome integrity. Interestingly, recent publications have shown that UvrD, Rep, Mfd and the single-stranded DNA-binding protein SSB are all important for R-loop homeostasis via unknown mechanisms (*Wimberly et al., 2013*; *Wolak et al., 2020*).

Although the function of PcrA/UvrD in transcription is still unclear, details are beginning to emerge about the interface formed between the helicase and RNAP which may provide some important clues. Crosslinking experiments revealed contacts between the 1B and 2B subdomains of the helicase and the β flap tip of the RNAP β subunit, as well as the N-terminal region of the RNAP β′ subunit (*Epshtein et al., 2014*). The extreme C-terminal domain (CTD) of PcrA, which is disordered or was removed to aid crystallisation in the available structures of this helicase, also plays a critical role in binding RNAP but its binding site on RNAP is unknown (*Gwynn et al., 2013*). A structure of the isolated CTD shows that it adopts a Tudor fold, which is connected to the main body of the helicase via a long, apparently disordered, linker (*Sanders et al., 2017*). This is intriguing, because the canonical TC-NER factor Mfd also employs a Tudor domain for interaction with RNAP (*Deaconescu et al., 2006*).

In this work, we find that PcrA interacts tightly with a transcription elongation complex in vitro in a manner dependent on the CTD. We show that (unlike Mfd) the CTD binds to a lineage-specific insertion domain within the β subunit of RNAP, while the main body of the helicase surrounds the RNA and DNA exit channels. The CTD engages with an exposed beta-hairpin which defines a novel helicase interaction motif also found in other PcrA partners including UvrB. We further show that PcrA efficiently unwinds DNA:RNA hybrids in vitro and that blocking PcrA activity in vivo, by either overexpressing a dominant negative mutant or by preventing PcrA-RNAP association, results in increased R-loop levels. On the basis of these observations, we develop a model for PcrA as an R-loop surveillance helicase, where it acts to unwind DNA:RNA hybrids and minimise the potentially toxic effects of transcription on DNA replication and repair.

## Results

### PcrA binds tightly to a transcription elongation complex using its C-terminal Tudor domain

Interaction between PcrA/UvrD and RNAP is well-documented in the literature, but little is known about which nucleic acid scaffolds or transcription states are the most favourable for association. In *E. coli*, a complex between RNAP and UvrD was reported to be stable enough to coelute during

size-exclusion chromatography (SEC) (*Epshtein et al., 2014*). In the *B. subtilis* system, although purified PcrA and RNAP do clearly interact in the absence of nucleic acids, we found that a DNA/RNA scaffold which mimics an elongation complex was required to form a sufficiently stable complex to resolve using SEC (*Figure 1—figure supplement 1A–B*). To rule out the possibility that the enhanced interaction was due simply to the helicase binding to the DNA, the complex formation was analysed further by electrophoretic mobility shift assays (EMSA) (*Figure 1—figure supplement 1C*). PcrA efficiently supershifted the transcription elongation complex (TEC; a pre-formed complex between RNAP and a DNA/RNA scaffold), whereas it was barely able to bind to the scaffold alone. Based upon EMSA, the apparent dissociation constant was ~500 nM and the interaction was dependent on the C-terminal Tudor domain of PcrA (CTD; *Figure 1A*). Removal of the CTD almost eliminated complex formation at concentrations of up to 1.5 µM of PcrA. Moreover, the purified CTD alone supershifted the TEC efficiently, albeit to a lower position in the gel. Although the UvrD CTD has been shown to possess a weak DNA binding activity (*Kawale and Burmann, 2020*), in our hands the PcrA CTD does not detectably bind to either single- or double-stranded DNA (Kd >> 5 µM) (*Sanders et al., 2017*). Therefore, this result further supports the idea that the PcrA-TEC interaction (at least with this scaffold) is substantially mediated by protein-protein interactions. Additional support for this assertion was provided by EMSA supershift experiments in which the structure of the scaffold was varied (*Figure 1B*). PcrA efficiently shifted the TEC regardless of the presence or length of the upstream and downstream DNA in the scaffold.

Competition experiments showed that full-length PcrA and free CTD competed for TEC association, and there was no evidence for a ternary complex formed between PcrA, the free CTD and the TEC (*Figure 1—figure supplement 1D*). The addition of unlabelled CTD caused a decrease in the levels of fluorescently labelled PcrA bound to the TEC rather than a further supershift. This strongly suggests that full length PcrA and the CTD bind to the same (unknown) surface of RNAP and is consistent with the idea that the TEC recruits a single PcrA molecule. Note that the ability of the free CTD to outcompete full-length PcrA for RNAP association is relevant for in vivo experiments presented below.

## The C-terminal domain of PcrA interacts with the lineage-specific insertion domain 1 (SI1) of the β subunit of RNAP

Crosslinking of UvrD to RNAP has shown that the core helicase interacts with the RNAP β-flap tip and the N-terminal region of the β′ subunit in bacteria, but no information about the CTD-interaction domain was obtained (*Epshtein et al., 2014*). Therefore, we pursued a complementary approach in the *B. subtilis* system, using hydrogen-deuterium exchange coupled to mass spectrometry (HDX-MS) to study the complex formed between the free CTD and RNAP. This technique allows one to measure the rate of exchange of amide hydrogen atoms with deuterium atoms in the solvent. This rate is affected by hydrogen bonding, surface accessibility, protein dynamics and changes in the presence and absence of partner proteins and can therefore be used to probe potential binding interfaces or conformational changes.

We obtained good sequence coverage and redundancy for both the CTD and the major subunits of RNAP (*Table 1*). The experiment was first validated by analysing the protection pattern on the CTD. As expected, data for the CTD in the CTD-RNAP complex showed significant protection compared to the CTD alone (*Figure 1C*). To visualise the results, a homology model of the Tudor domain of *B. subtilis* PcrA was created from the structure of the closely related *G. stearothermophilus* protein (*Sanders et al., 2017*; *Figure 1D*). The areas of protection, which indicate the binding site location, overlap with residues that have been shown to be important for the interaction between PcrA and RNAP through site-directed mutagenesis studies (*Sanders et al., 2017*). Moreover, it is mostly localised on the face of the Tudor domain that is typically expected to interact with partner proteins (*Liu et al., 2015*; *Westblade et al., 2010*). This agreement with previous studies provided confidence that the HDX-MS data was reporting correctly on the interface formed between PcrA and RNAP.

We next searched for protected peptides in the different RNAP subunits. It should be noted that, given the large size of RNAP (374 kDa) compared to the tiny Tudor domain (9.3 kDa), we did not expect to observe any protection for the large majority of the RNAP sequence if the stoichiometry of interaction is 1:1. In accordance with this expectation, only a small region of the β subunit of RNAP at approximately amino-acid position 300 showed significant protection (*Figure 2A*, with HDX

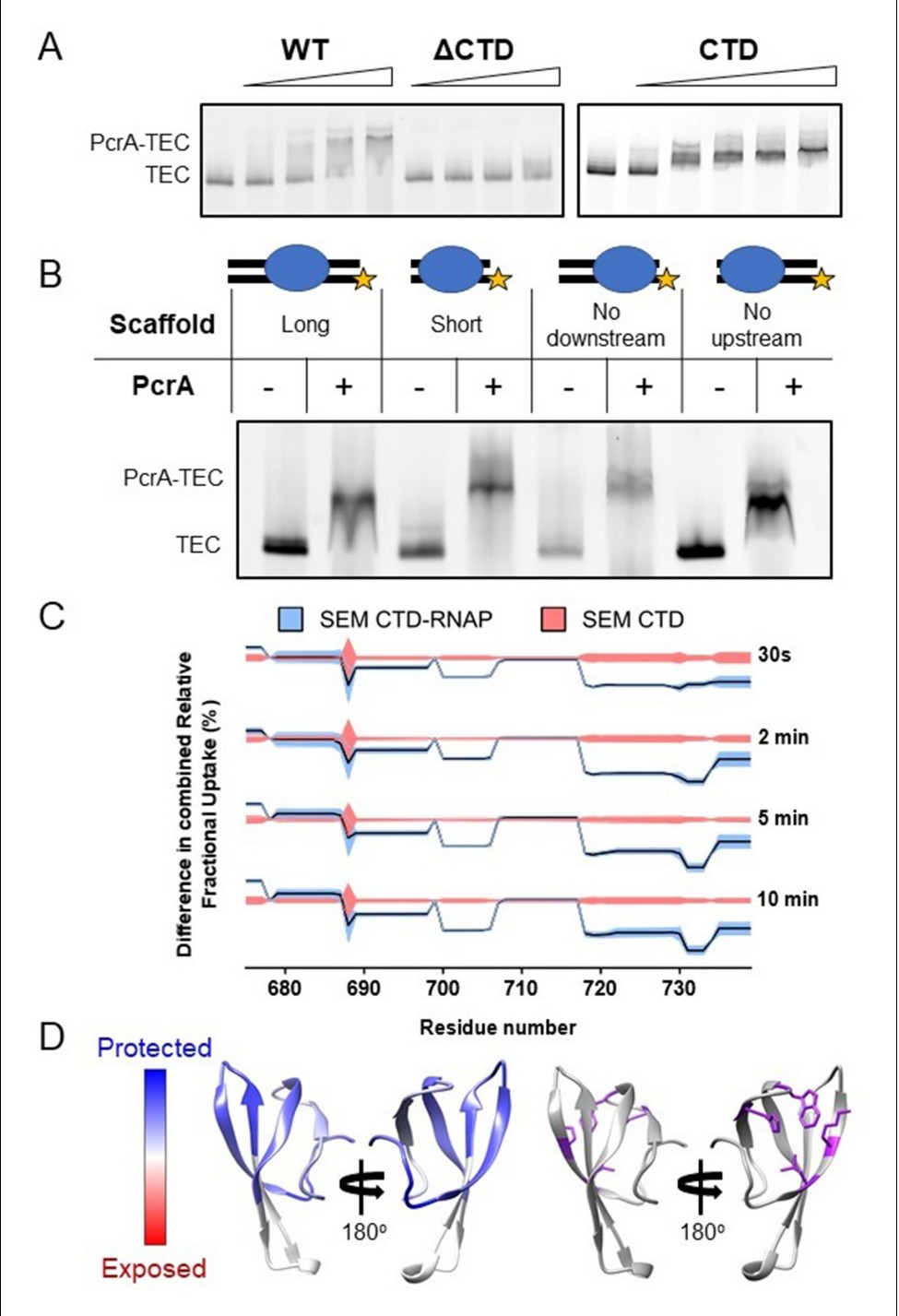

**Figure 1.** Interactions between PcrA and a transcription elongation complex are mediated by protein-protein interactions involving the PcrA-CTD. (**A**) EMSA supershift assays to monitor association of PcrA with a TEC. The PcrA-CTD is necessary and sufficient for stable formation of the PcrA-RNAP complex. WT PcrA and ΔCTD PcrA were titrated from 0.25 µM to 1.5 µM. The PcrA-CTD was titrated from 0.5 µM to 3 µM. (**B**) EMSA supershift assay showing that binding of PcrA is not dependent on the presence of upstream or downstream DNA in the TEC. The star indicates the position of the fluorescent label at the 5′ end of the template strand of the scaffold. PcrA was used at 1 µM. The oligonucleotides used to assemble the scaffolds are shown in *Table 6*. (**C**) Relative HDX measured for the PcrA-CTD in the CTD-RNAP complex compared to CTD alone. The black line shows the differential relative uptake and the pink and blue shadowing, the SEM of the CTD and CTD-RNAP conditions, respectively. The four offset traces show different exchange times. Negative uptake values over the CTD baseline

*Figure 1 continued on next page*

*Figure 1 continued*

with non-overlapping shadowing show protected amino acid regions on the CTD when it is in complex with RNAP. Note the key regions of the CTD (aa ~690–705 and aa ~720-Ct) are protected upon binding RNAP. (D) Left - homology model of the PcrA-CTD showing regions protected from HDX in the complex with RNAP (dark blue). Right - amino acids known to be important for interaction with RNAP (purple residues).

The online version of this article includes the following figure supplement(s) for figure 1:

**Figure supplement 1.** Interactions between PcrA and a transcription elongation complex are mediated by protein-protein interactions involving the PcrA-CTD.

protection plots for the rest of RNAP β and the other RNAP subunits in *Figure 2—figure supplement 1*). Comparison of the HDX at different timepoints showed that protection of this region increased with time after exposure to $D_2O$, in agreement with a protein-protein interaction site. Fortunately, this region of interest was well covered by the LC-MS/MS analysis with overlapping peptides, and the smallest part of the polypeptide protected was the seven-residue sequence PETGEIL (*Figure 2A and B*). Intriguingly, despite the ubiquity of UvrD/PcrA orthologues in the bacterial domain and the strong conservation of the CTD sequence, the protected region in RNAP belongs to a lineage-specific insertion domain (SI1, also referred to as βi5 [*Lane and Darst, 2010*]) which differs significantly between organisms in terms of the length, sequence and precise location within the primary structure. Nevertheless, a nine-residue sequence motif (including the seven amino acids of interest) is present in most bacterial RpoB sequences despite variation in its exact position. As an example, primary structure diagrams for the *B. subtilis* and *E. coli* RpoB subunits are shown in *Figure 2B* with complete aligned sequences for the β2 and SI1 regions shown in *Figure 2—figure supplement 2*. Despite being quite divergent compared to better-conserved parts of RpoB, the two SI1 domains contain a very similar peptide motif (VDPETGEIL and IDESTGELI respectively). Although the motifs are located in different parts of the SI1 domain primary structure they occupy a broadly similar space within the global architecture of RNAP (*Figure 2C*). In *E. coli* the motif adopts a beta hairpin structure, whereas this region is locally disordered in the *B. subtilis* TEC cryoEM structure (*Newing et al., 2020*; *Pei et al., 2020*). The variability in the SI1 domain would make it extremely challenging to appreciate the conservation of this motif across different species from primary sequence alone.

Alignment of the beta-hairpin motif from *B. subtilis*-like SI1 domains demonstrated its high conservation (shown in Weblogo format in *Figure 2C*). The consensus sequence obtained from *E. coli*-like SI1 domains is very similar, consistent with the idea that they are in fact the same motif (*Figure 2C*). The most conserved and striking feature of this putative helicase interaction motif is a TGE triad which occupies the tip of a loop between two beta strands. Indeed, the glutamic acid (E310 in *B. subtilis* RpoB) is absolutely conserved across all aligned RpoB sequences. To experimentally validate the motif, we purified the β subunit of *B. subtilis* RNAP with and without substitutions at the E310 position. In vitro assays showed that substitution of glutamic acid with either alanine or lysine at position 310 severely disrupts pulldown of purified RpoB by PcrA (*Figure 2D*), confirming

**Table 1.** Sequence coverage and redundancy for the proteins analysed by HDX-MS.

| Protein | CTD-RNAP complex | | PcrA-RNAP complex | |
| --- | --- | --- | --- | --- |
| | Sequence coverage (%) | Sequence redundancy | Sequence coverage (%) | Sequence redundancy |
| CTD/ PcrA | 87.2 | 3.61 | 78.5 | 2.38 |
| α | 93.9 | 3.48 | 89.2 | 2.59 |
| β | 83.7 | 2.84 | 84.9 | 2.24 |
| β′ | 88.7 | 2.58 | 80.3 | 2.05 |
| δ | 75.7 | 1.79 | 17.3 | 1.53 |
| ε | 100 | 4.26 | 91.3 | 2.38 |
| ω | - | - | - | - |
| σ$^A$ | 42.6 | 1.29 | - | - |

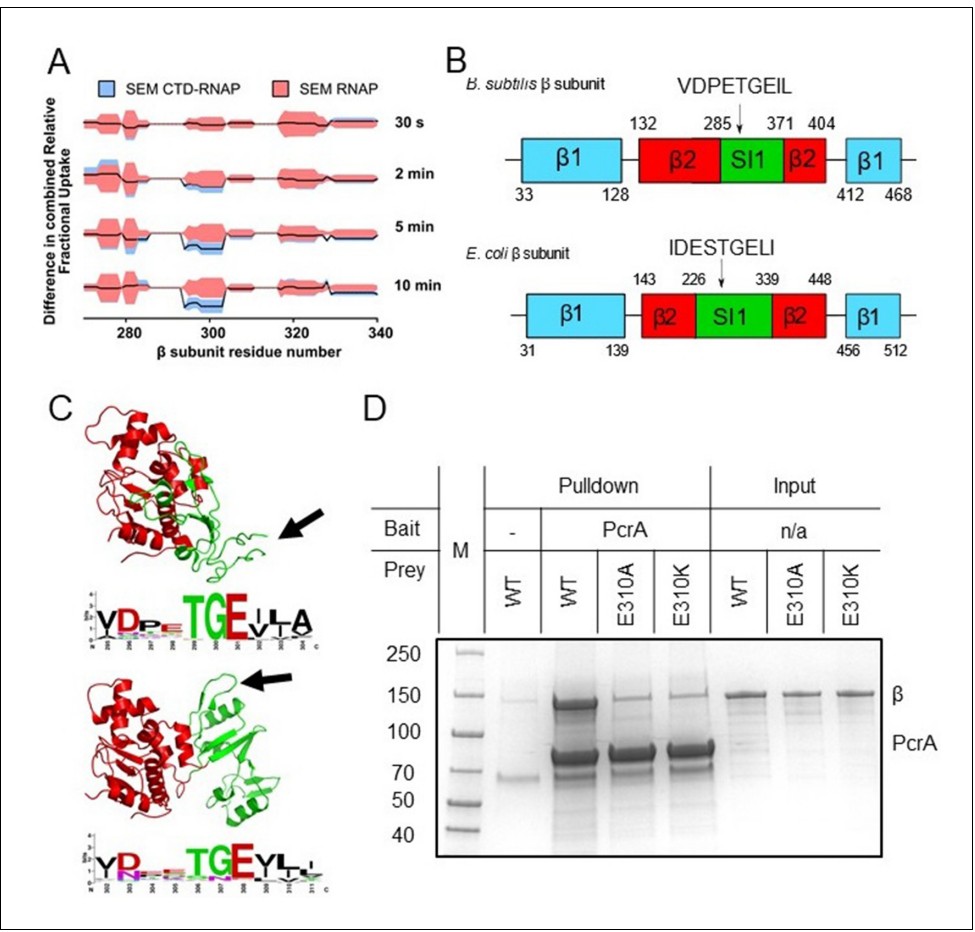

**Figure 2.** The PcrA-CTD binds to a conserved motif in the SI1 domain of RNAP. (**A**) Relative HDX measured for a region of the RNAP β subunit (residue numbers on x axis) within the CTD-RNAP complex (blue) compared to RNAP alone (red). A small region of RpoB (at amino-acid positions around ~300) becomes significantly protected by interaction with the PcrA CTD as the exchange time becomes longer. (**B**) The protected region maps to a conserved motif in the SI1 domain of *B. subtilis* RpoB. This region is organised differently in *E. coli* RpoB, but the same conserved amino acid motif appears in a slightly different position in the structure (black arrow). (**C**) Structure of the *B.subtilis* (upper panel) and *E. coli* (lower panel) β2 (red) - SI1(green) domains indicating the beta-loop structure containing a putative interaction motif at the tip (black arrows). This sequence is well-conserved in bacterial RNA polymerases and the consensus sequence is shown in weblogo format beneath each structure. (**D**) In vitro pulldown of RpoB using PcrA as a bait (see Materials and methods for details). Mutation of the conserved glutamate (E301) in the putative helicase interaction motif dramatically reduces RpoB pulldown.

The online version of this article includes the following figure supplement(s) for figure 2:

**Figure supplement 1.** HDX protection plots for the remaining RNAP subunits in the CTD-RNAP experiment.

**Figure supplement 2.** The CTD-interacting motif is located in different regions of the SI1 domain among landmark organisms.

**Figure supplement 3.** Mutations to E301 do not alter the overall structure of the β subunit of RNAP.

the importance of the SI1 motif for the interaction. Lack of interaction with the E308 mutants was not due to global mis-folding because both mutant proteins had equivalent CD spectra to wild type (*Figure 2—figure supplement 3*).

## The helicase interaction motif is found in several other PcrA interaction partners including UvrB

In common with other SF1 helicases, PcrA/UvrD interacts with many different partner proteins explaining its recruitment to different DNA transactions and the resulting multi-functionality of this helicase (*Gilhooly, 2013*). To investigate whether the helicase interaction motif we had identified in

RpoB played a wider role in mediating PcrA interactions, we searched for it within the *Bacillus subtilis* proteome using PROSITE with the query [VIL]-D-X-X-T-G-E-[VIL]-[VILT], where X is any amino acid and the square brackets indicate ambiguity. This search retrieved seven proteins (only ~0.3 are expected by chance), four of which (RpoB, UvrB, YxaL, and YwhK) are known to interact with PcrA (*Figure 3A*). Importantly, the structure of UvrB and high-confidence homology models of YxaL and YwhK all show that, as is the case for the beta subunit of *E. coli* RNAP, the conserved TGE triad is

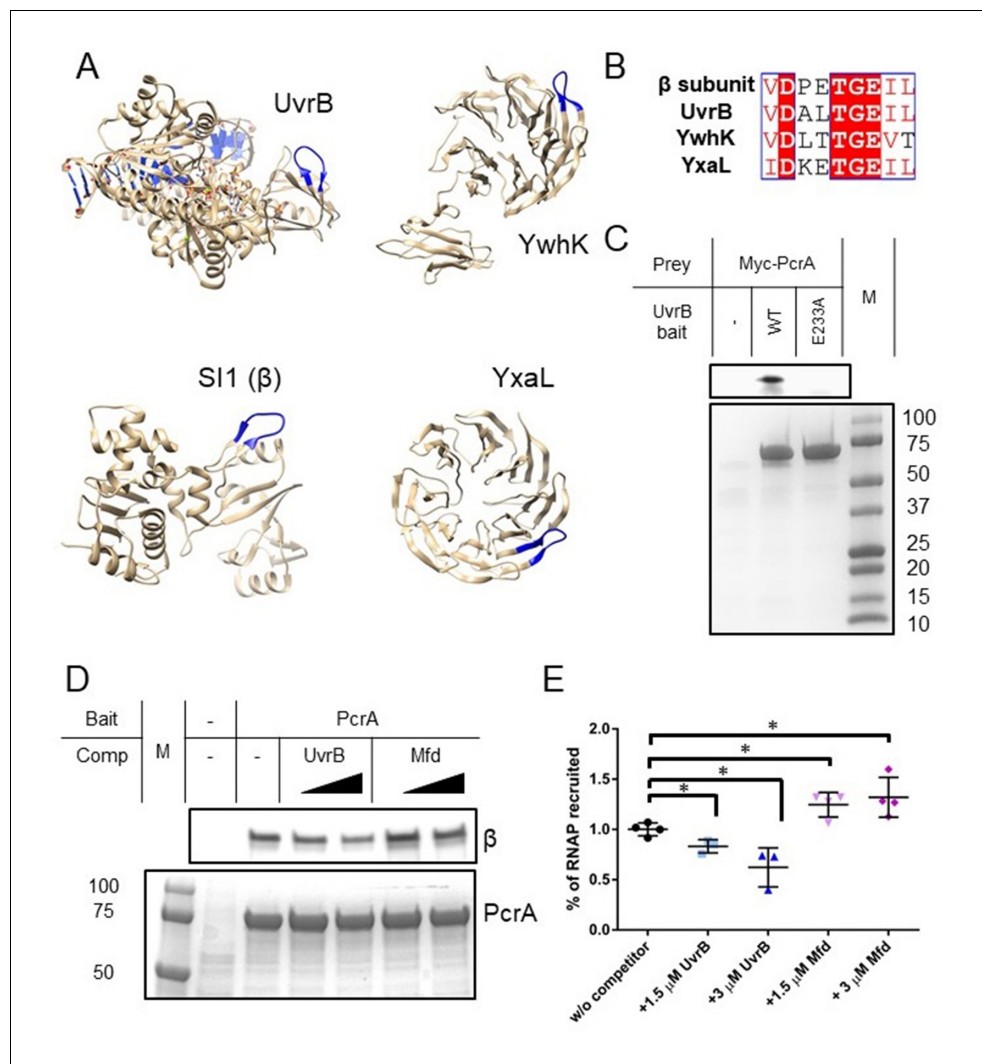

**Figure 3.** Many PcrA partner proteins contain the helicase interaction motif. (A) Putative helicase interaction motif (blue) in known PcrA interaction partners from *B. subtilis*. A beta hairpin structure (blue) is formed by the interaction motif in each of the proteins. Two of the structures are homology models as indicated in the text. (B) Sequences of putative helicase interaction motifs in four known PcrA partner proteins. (C) Pulldown of Myc-tagged PcrA from *B. subtilis* cell extracts using UvrB as bait (for details see the Materials and methods). Mutation of the conserved glutamate (E233A) in the putative helicase interaction motif dramatically reduces PcrA pulldown. PcrA was detected using an anti-Myc antibody (upper gel). Equivalent loading of WT and mutant UvrB was confirmed by Coomassie staining (lower gel). (D) and (E) Pulldown of RNAP from *B. subtilis* cell extracts using biotinylated PcrA as bait. Where indicated the prey was supplemented with purified UvrB or Mfd. Free UvrB, but not Mfd, competes for the interaction site formed between PcrA and RNA polymerase. Error bars show the SEM of three independent experiments. Two-tailed Student's t test determined statistical significance (*p value < 0.05).

The online version of this article includes the following figure supplement(s) for figure 3:

**Figure supplement 1.** The E233A mutation does not affect the ATPase activity of UvrB.

**Figure supplement 2.** The CTD interacts with UvrB domain two and close to the damaged DNA site.

located at the tip of a surface loop flanked by β sheets (*Figure 3B*). There is currently no evidence for interaction between PcrA and the other three proteins (QueA, RplX, and YtzB) which may now be considered as putative PcrA interactors. When this same motif search was applied more broadly across the entire SwissProt database it returned only 607 sequences, of which 326 and 42 were bacterial RpoB and UvrB sequences respectively, demonstrating its very high specificity for returning PcrA interactors.

YxaL and YwhK are β propeller proteins that have been identified as PcrA interactors using yeast two-hybrid and biochemical assays (*Noirot-Gros et al., 2002*). The fragment of YxaL found to interact with PcrA in the yeast two-hybrid assay is consistent with the location of the newly identified motif. These two proteins are found in a very limited subset of bacterial proteomes and their cellular function is obscure. Therefore, we next decided to focus on the interaction of PcrA with UvrB; a ubiquitous Superfamily two helicase that is a key component of the nucleotide excision repair apparatus (*Kisker et al., 2015*). Importantly, UvrB has already been shown to bind to the C-terminal domain of *E. coli* UvrD (*Manelyte et al., 2009*) and this interaction is, therefore, potentially analogous to that formed between the CTD and RNAP β. To validate the predicted PcrA binding site on UvrB, we performed pulldown experiments using his-tagged UvrB as bait and myc-tagged PcrA in *B. subtilis* extracts as prey (*Figure 3C*). These experiments confirmed the expected interaction between PcrA and wild-type UvrB protein. Furthermore, and in agreement with our hypothesis, substitution of the glutamic acid in the UvrB TGE motif (E233) severely disrupted the interaction with PcrA while having no effect on the DNA-stimulated ATPase activity of UvrB (*Figure 3—figure supplement 1*).

Inspection of the domain architecture of UvrB reveals that the PcrA interaction motif is located within domain 2, which also interacts with UvrA (*Pakotiprapha et al., 2012*; *Truglio et al., 2004*; *Figure 3—figure supplement 2A*). However, because amino acid substitutions reported to disrupt UvrA-UvrB interaction are not within the motif itself, binding of UvrA to UvrB would not necessarily preclude binding of PcrA/UvrD. Interestingly, a high scoring docking model for the interaction between the PcrA-CTD and UvrB using HADDOCK (*van Zundert et al., 2016*) suggests that the complex is stabilised by an electrostatic interaction between UvrB E233 and the PcrA K727 residue, which is known to be critical for interaction between PcrA and RNAP (*Figure 3—figure supplement 2B*). The docked complex shows how PcrA might be recruited to sites of damage after the damaged oligonucleotide has been nicked, as DNA near the docking site is accessible. A conservation logo for the PcrA interaction site on UvrB was created from reference proteomes (*Figure 3—figure supplement 2C*). The consensus sequence is highly similar to that obtained for the RNAP interaction although, interestingly, the glutamate residue shows somewhat greater variability.

Our results predict that the interaction between the PcrA C-terminal Tudor domain and UvrB is equivalent in molecular terms to its interaction with RNA polymerase. A corollary is that UvrB and RNA polymerase might be engaged in a simple competition for the Tudor domain of PcrA. To test this, we performed pulldown of RNA polymerase using PcrA as a bait and titrated the cell extract (i.e. the prey proteins) with increasing concentrations of purified recombinant UvrB (*Figure 3D–E*). As expected, introduction of UvrB significantly reduced the amount of RNA polymerase that was recovered from the cell extract. As has been noted previously, other RNAP interactors also bind to the RpoB subunit via Tudor domains, including the transcription-repair coupling factor Mfd. Interestingly, using the same pulldown assay, titration of free Mfd did not diminish the yield of RNAP (*Figure 3D–E*), despite biotinylated Mfd pulling down RNAP subunits under similar conditions (*Table 2*). This suggests that, although both proteins contain Tudor domains for interaction with RNAP, PcrA, and Mfd bind at different physical locations and without competing. This is in agreement with recently-published observations for the *E. coli* system (*Kawale and Burmann, 2020*), as well as the lack of protection of the expected Mfd-interacting region of RpoB (the β1 domain) that we observe in our HDX experiments (*Figure 2A* and *Figure 2—figure supplement 1A*; *Smith and Savery, 2005*; *Westblade et al., 2010*).

## The helicase core of RNAP interacts with RNAP near the RNA and DNA exit channels

Despite the apparently critical role played by the CTD in RNAP interactions, previous work in live *E. coli* and *B. subtilis* cells has found at most mild phenotypes associated with deletion of the C-terminal domain of UvrD and PcrA, respectively (*Manelyte et al., 2009*; *Merrikh et al., 2015*; *Sanders et al., 2017*). This led us to consider the possibility that the N-terminal region of PcrA,

**Table 2.** Mfd pulls down RNAP subunits from *Bacillus subtilis* cell extracts.
Proteins enriched in the biotinylated Mfd bait condition compared to the no-bait control pulldown (see Materials and methods for details). Subunits of RNAP that are enriched in the Mfd pulldown are indicated in bold text. Accession refers to the UniProt accession code, GN refers to gene name and FC, to fold change.

| Accession | Description | LogFC |
|---|---|---|
| P37474 | Transcription-repair-coupling factor GN=mfd | 7.887 |
| O34863 | UvrABC system protein A GN=uvrA | 6.127 |
| O34628 | Uncharacterised protein YvlB GN=yvlB | 5.779 |
| Q795Q5 | Uncharacterised membrane protein YttA GN=yttA | 3.735 |
| O34942 | ATP-dependent DNA helicase RecG GN=recG | 3.304 |
| Q06796 | 50S ribosomal protein L11 GN=rplK | 3.269 |
| **P20429** | **DNA-directed RNA polymerase subunit alpha GN=rpoA** | **2.899** |
| O32006 | Resolvase homolog YokA GN=yokA | 2.898 |
| O07542 | UPF0342 protein YheA GN=yheA | 2.800 |
| **O35011** | **DNA-directed RNA polymerase subunit omega GN=rpoZ** | **2.780** |
| P39592 | Uncharacterised HTH-type transcriptional regulator YwbI GN=ywbI | 2.667 |
| Q08792 | Uncharacterised HTH-type transcriptional regulator YcxD GN=ycxD | 2.654 |
| O34949 | Uncharacterised HTH-type transcriptional regulator YkoM GN=ykoM | 2.600 |
| O34381 | HTH-type transcriptional regulator PksA GN=pksA | 2.512 |

including the helicase core, also contributes significantly to the interaction with RNA polymerase. This view is consistent with crosslinking studies of UvrD-RNAP which implicated the 1B and 2B subdomains of UvrD in interactions with the RNAP β-flap tip and the N-terminal region of the β′ subunit (*Epshtein et al., 2014*). Note that these structural features surround the RNA and DNA exit channels.

We did not pursue HDX experiments between PcrA-ΔCTD and RNAP because our previous work had shown that this interaction was barely detected by pulldown/proteomics experiments in vitro (*Gwynn et al., 2013*; *Sanders et al., 2017*) even with the favoured TEC substrate (Kd > 1.5 μM; *Figure 1A*). Instead, to compare with data for the CTD-RNAP complex, we studied interaction between full-length PcrA and RNAP, obtaining good coverage and redundancy values for the various constituent polypeptides (*Table 1*). PcrA showed HDX protection in two main regions upon interaction with RNAP: the 2A domain and the CTD (blue and magenta highlights respectively; *Figure 4A* and *Figure 4—figure supplement 1A*). Note that peptides from the long flexible linker between these two regions were not detected above the confidence threshold and so we have no data for that part of the protein. The protection pattern was visualised on a homology model of the active DNA bound state of PcrA (*Figure 4B*; *Lee and Yang, 2006*; *Ordabayev et al., 2019*; *Velankar et al., 1999*), in which the CTD is disordered. Most of the protected regions outside of the CTD prominently cluster in the 2A domain and on the side of the helicase which also binds to DNA. The crosslinking positions identified in domains 1B and 2B in previous work are on the same DNA-facing side of the helicase and are therefore broadly consistent with these data (*Epshtein et al., 2014*).

The HDX protection afforded to the RNAP subunits in the presence of full length PcrA was more extensive than in the presence of the CTD, and identified important regions in the two largest subunits, β and β′ (butterfly plots for the rest of the subunits are shown in *Figure 5—figure supplement 1*). In agreement with our data for the CTD-RNAP interaction, the SI1 region showed protection that we therefore assign to the interaction with the PcrA Tudor domain (*Figure 5A*, green shadow and *Figure 6B*). The other protected region within the β subunit corresponds to the β flap tip, as identified by crosslinking in the *E. coli* system (*Epshtein et al., 2014*; *Figure 4—figure supplement 1B*). The β′ subunit also showed two protected clusters in the N-terminal and central regions (*Figure 5B* and *Figure 4—figure supplement 1C*). The recent structure of the *B. subtilis* transcription

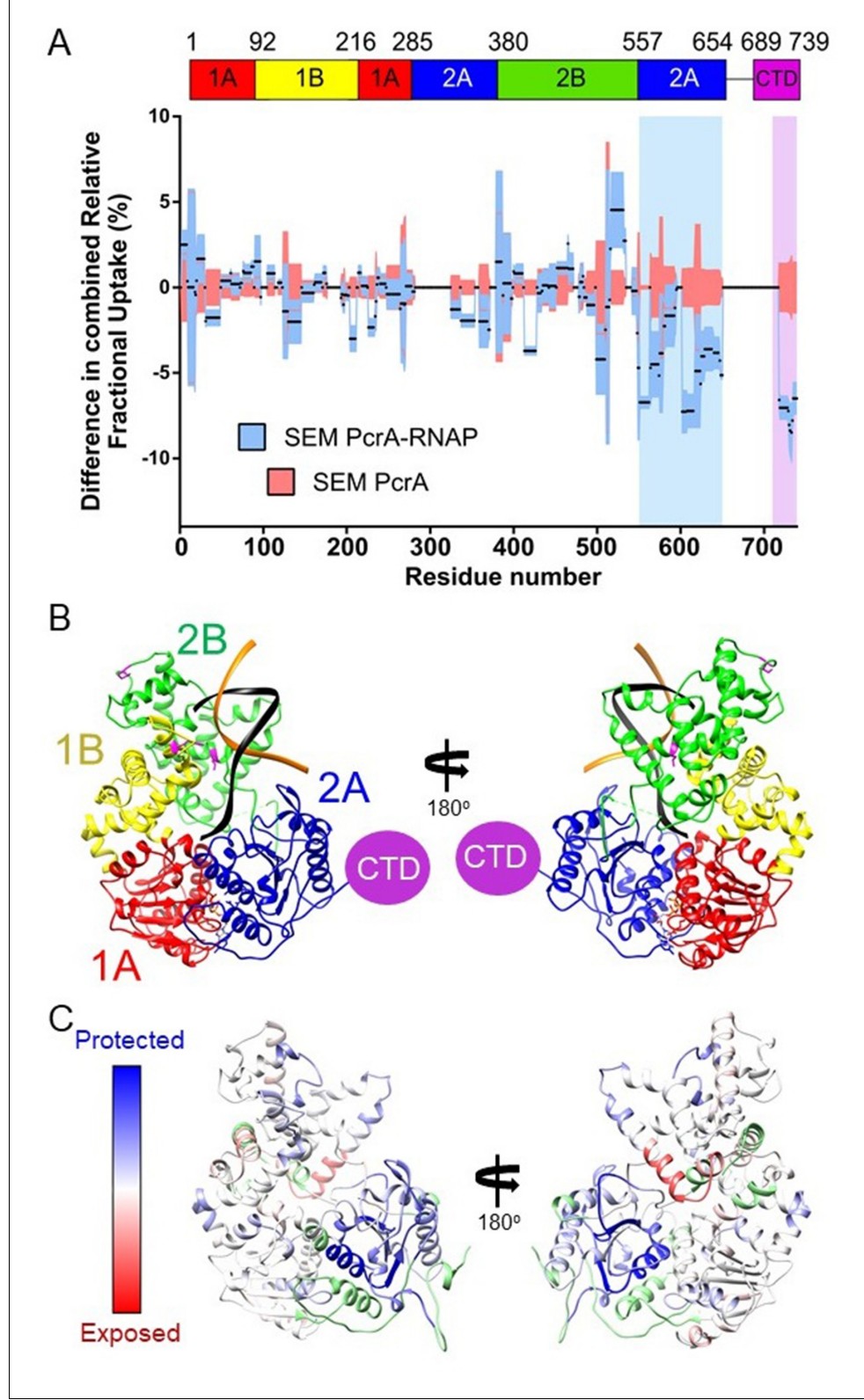

**Figure 4.** The PcrA helicase core is protected by interaction with RNA polymerase. (**A**) Relative HDX measured for full length PcrA (residue numbers on x axis) in the PcrA-RNAP complex (blue) compared to PcrA alone (red). The magenta rectangle highlights strong protection of the CTD afforded by interaction with RNAP as expected based on results presented earlier in this manuscript. The blue rectangle highlights a second region of strong protection

*Figure 4 continued on next page*

*Figure 4 continued*

within the 2A domain of PcrA. (**B**) The PcrA helicase core (homology model from PDB: 3PJR) showing domain organisation. The DNA substrate is shown in black and orange and the CTD (which is disordered in this structure) is indicated as a purple circle. (**C**) The same structure showing the mapping of the HDX-protection data (bottom; blue indicates protection in the complex, red indicates exposure and green indicates a lack of data). Note that the HDX-protection data maps largely to one face of the helicase within domains 2A and, to a lesser extent, 2B.

The online version of this article includes the following figure supplement(s) for figure 4:

**Figure supplement 1.** Deuterium uptake dynamics for regions of PcrA and RNAP that are significantly protected in the PcrA-RNAP complex.

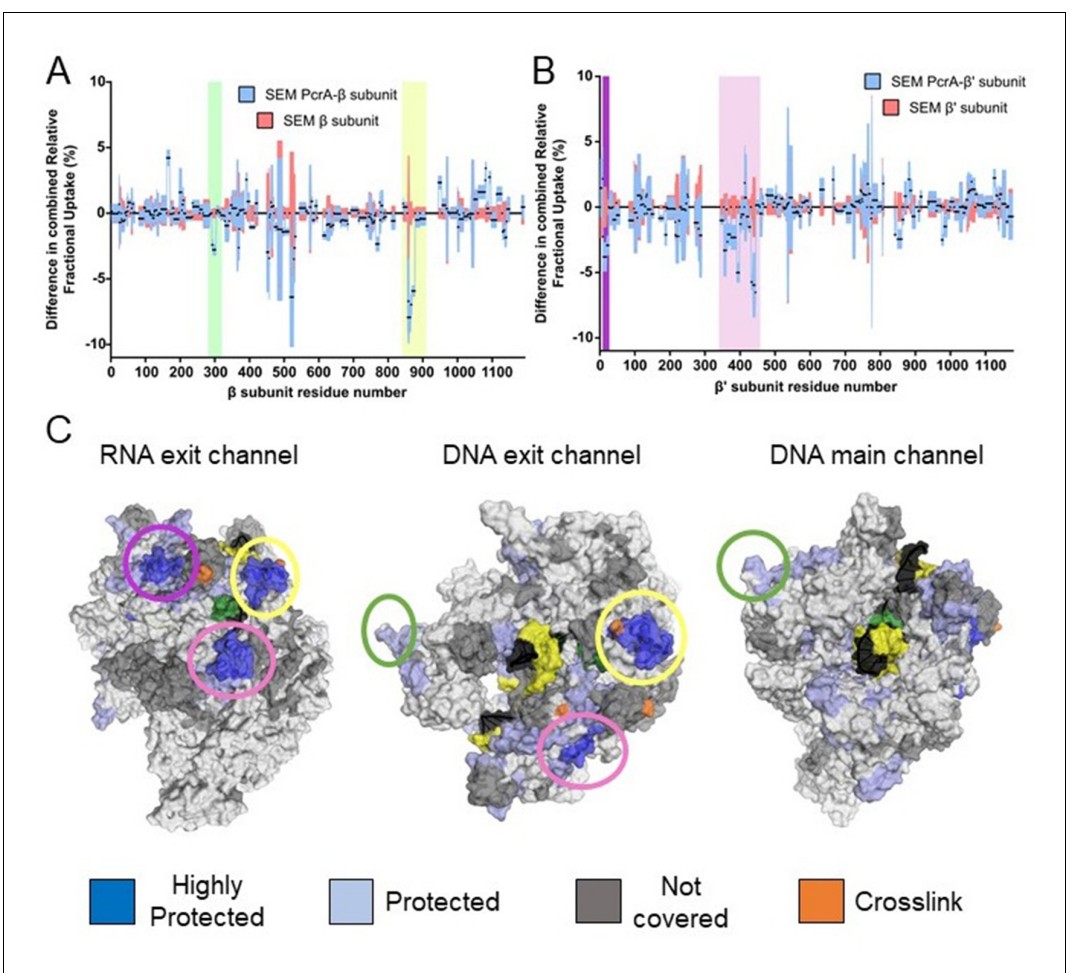

**Figure 5.** The PcrA helicase core binds to RNAP close to the RNA and DNA exit channel. (**A**) Differential protection plot for the β subunit of RNAP showing the differential relative uptake at 10 min of exposure to deuterium for a PcrA-RNAP complex (blue) compared to RNAP alone (red). Green and yellow rectangles highlight regions of RpoB that are significantly protected in the complex state. (**B**) Differential protection plot for the β′ subunit of RNAP showing the differential relative uptake at 10 min of exposure to deuterium for a PcrA-RNAP complex compared to RNAP alone. Purple and pink rectangles highlight regions of RpoC that are significantly protected in the complex state. (**C**) Three views of the *B. subtilis* TEC (PDB: 6WVJ) showing regions protected by interaction with PcrA. DNA is shown in yellow and black. RNA is green. Coloured circles correspond to the protected regions highlighted in panels A and B. The green circle highlights the helicase interaction motif already identified as the site of binding of the PcrA-CTD. Note that the other major protection sites surround the DNA/RNA exit channels. The crosslinking sites (orange) are from reference (*Epshtein et al., 2014*).

The online version of this article includes the following figure supplement(s) for figure 5:

**Figure supplement 1.** HDX protection data for the remaining RNAP subunits in the PcrA-RNAP interaction experiment.

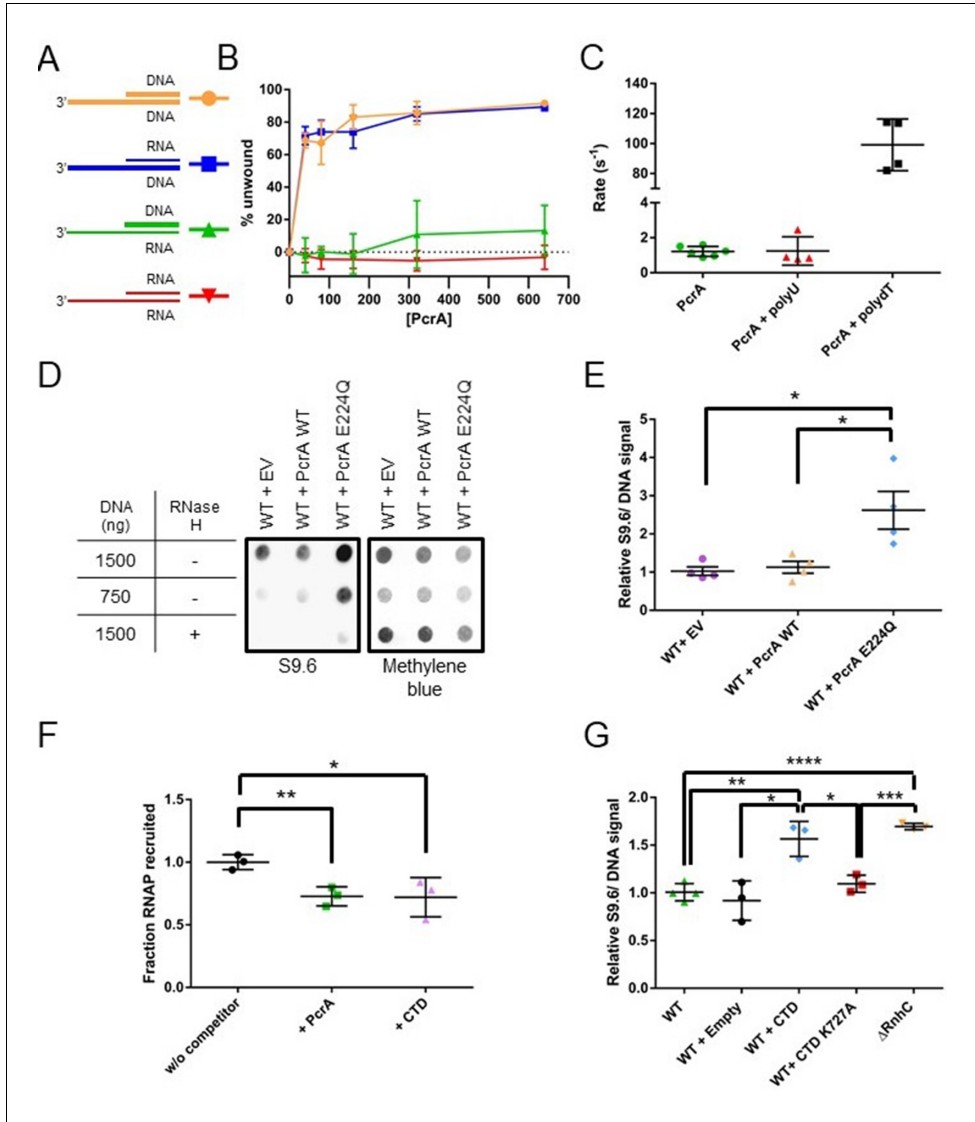

**Figure 6.** PcrA unwinds DNA-RNA hybrids in vitro and supresses R-loops in vivo (A) DNA and RNA substrates used for helicase assays. Thick lines represent DNA strands and thin lines, RNA strands. The oligonucleotides used to form these substrates are shown in *Table 7*. (B) Quantification of unwinding as a function of PcrA concentration (in nM) for the 3′-tailed substrates shown in panel A. The substrate is only efficiently unwound if the longer of the two nucleic acids strands is DNA. Error bars show the standard deviation of at least three independent experiments. (C) The ATPase activity of PcrA is strongly stimulated by single-stranded DNA but not single-stranded RNA. Error bars show the standard deviation of at least three independent experiments. (D) Anti R-loop antibody (S9.6) dot blot for nucleic acid samples purified from three strains of *B. subtilis*. These strains contain an integrated expression cassette for either wild-type PcrA or a dominant negative form of PcrA (E224Q). The control strain (EV) contains an integrated but empty expression cassette. The S9.6 signal is normalised using methylene blue as a stain for all DNA. Note the high S9.6 signal for the strain expressing PcrA E224Q. (E) Quantification of four independent repeats of the experiment shown in (c). Error bars show the SEM. Expression of a dominant negative form of PcrA increases R-loop content (relative to DNA) in *B. subtilis* by ~2.5 fold. (F) Quantification of pulldown experiments of RNAP from *B. subtilis* cell extracts using biotinylated PcrA as bait and supplemented with purified PcrA WT or CTD. Addition of the CTD competes with WT PcrA to bind to RNAP. Error bars show the SEM of three independent repeats. (G) Relative R-loop levels in strains of *B. subtilis* expressing free CTD, a CTD mutant that interacts weakly with RNAP, or with a control expression cassette. A Δ*rnhC* strain is shown as a control for elevated R-loop levels. Error bars show the SEM of at least three independent experiments. In all panels, the statistical significance was determined using two-tailed Student's t test (*p value < 0.05, **p value < 0.01, ***p value < 0.001, ****p value < 0.0001).

*Figure 6 continued on next page*

*Figure 6 continued*

The online version of this article includes the following figure supplement(s) for figure 6:

**Figure supplement 1.** PcrA requires a 3′-ssDNA tail to unwind DNA duplexes.

**Figure supplement 2.** Overexpression of PcrA E224Q causes growth defects in WT and Δ*mfd B. subtilis*.

**Figure supplement 3.** Overexpression of the PcrA CTD in *B. subtilis* and deletion of *uvrD* in *E. coli* increase R-loop levels in the cell.

elongation complex allowed us to visualise the results (*Newing et al., 2020*; *Figure 5C*) and reveals that protected regions of RNAP surround the RNA and DNA exit channels. Unfortunately, the zinc finger domain of the β′ subunit was not covered by our dataset but, given its close proximity to the protected regions we have observed, as well as the crosslinking analysis of the *E. coli* RNAP-UvrD complex (*Epshtein et al., 2014*), it is likely to be involved in the interaction too. Taken together, our data suggested to us that the PcrA helicase might be acting close to the RNA and DNA exit channels and are consistent with our earlier conclusion that Mfd and PcrA bind to different parts of RNAP.

## PcrA is able to unwind DNA-RNA hybrids by translocating on ssDNA

It is well established that PcrA/UvrD is a 3′>5′ ssDNA translocase and helicase that binds preferentially to 3′-ssDNA overhangs to initiate unwinding of flanking DNA duplexes. However, UvrD has also been shown to unwind DNA-RNA hybrids efficiently (*Matson, 1989*), although we are not aware of a cellular function having been ascribed to this activity. Moreover, no data is available for the enzyme's activity on RNA duplexes. Therefore, given the apparent recruitment of PcrA close to the RNA and DNA exit channels of RNAP, we next investigated the relative helicase activity displayed by PcrA against all possible combinations of a 3′-tailed duplex that can be formed by DNA and RNA oligonucleotides (*Figure 6A*). PcrA was able to efficiently unwind the two substrates in which the 3′-ssDNA tail was formed from DNA, including an all-DNA duplex and a DNA-RNA hybrid (*Figure 6B*). In contrast, the enzyme was unable to unwind either substrate containing a 3′-ssRNA tail (either an RNA-DNA hybrid or an RNA duplex), even at elevated concentrations. Unwinding of both the DNA duplex and the RNA-DNA hybrid was absolutely dependent on the presence of the 3′-ssDNA tail (*Figure 6—figure supplement 1A–B*), implying that it is initiated by enzymes binding to ssDNA and moving in the 3′>5′ direction. This idea was further corroborated by investigating the nucleic acid-dependence of the ATPase activity (*Figure 6C*). In the absence of DNA or RNA, the basal ATPase activity of PcrA was low (~1 s$^{-1}$). However, it was stimulated approximately 100-fold in the presence of poly(dT) single-stranded DNA. In contrast, poly(U) single-stranded RNA had no effect on the observed ATPase activity. Similar results were obtained with short DNA and RNA oligonucleotides containing mixed base sequences (data not shown). These experiments demonstrate that PcrA helicase can efficiently unwind DNA-RNA hybrids but can only do so by ATP-dependent translocation on the DNA strand. Together with the HDX experiments localising PcrA to the RNA and DNA exit channels, we hypothesised that PcrA might act to unwind R-loops formed during transcription.

## PcrA helicase suppresses R-loop levels in vivo

Both PcrA and UvrD have been shown to interact with RNAP and to alleviate replication-transcription conflicts (in *B. subtilis* and *E. coli*, respectively) (*Guy et al., 2009*; *Hawkins et al., 2019*; *Merrikh et al., 2015*). However, the precise role for this interaction remains unknown and study of the system is frustrated by the essentiality of both PcrA and RNA polymerase, making phenotypic analysis more complex. To overcome this difficulty, we first exploited a dominant negative mutant of PcrA/UvrD in which a glutamate in the Walker B motif is substituted to allow DNA and ATP binding, but not hydrolysis and concomitant DNA translocation (*Brosh and Matson, 1995*). Studies in many Additional Strand Catalytic E (ASCE)-class NTPases identify this residue as the catalytic base which accepts a proton from water, thereby promoting nucleophilic attack at the γ-phosphate and enabling ATP hydrolysis (*Thomsen and Berger, 2008*). Substitution of this glutamate residue with glutamine prevents ATP hydrolysis, trapping the enzyme in an ATP-bound or transition-like state (*Hirano and Hirano, 2004*; *Soultanas et al., 1999*).

Induction of PcrA-E224Q from an ectopic locus in otherwise wild-type cells caused severe growth defects which became most apparent as the cells entered exponential phase (*Figure 6—figure supplement 2A*). This defect was not apparent when wild type PcrA was expressed from the same locus and is therefore due to the E224Q mutation as opposed to the effects of overexpression per se or the elevated cellular PcrA concentration that results. Interestingly, delaying the induction of PcrA-E224Q until the onset of exponential phase (see Materials and methods and *Figure 6—figure supplement 2A* for details) resulted in a much smaller effect, presumably because a lower concentration of mutant PcrA had accumulated in early exponential phase. This may suggest that the toxic effect of dysfunctional PcrA is mainly felt during rapid growth. Removal of the CTD from the dominant negative mutant did not relieve the toxicity we observed. This could indicate that PcrA-E224Q lacking the CTD can still interact weakly with RNAP (as we have discussed above) or that the toxic effect we observe arises from another role of this multifunctional enzyme that does not require the CTD. We also tested whether removal of *mfd* (that has been shown to counteract PcrA essentiality [*Moreno-Del Alamo et al., 2020*]) could relieve the mutant's toxicity but found that the deletion had no substantial effects on the growth defect (*Figure 6—figure supplement 2B*).

To test our hypothesis that PcrA unwinds R-loops, we next measured DNA-RNA hybrid levels in these PcrA overexpression strains using the S9.6 antibody (*Boguslawski et al., 1986*). In this experiment, genomic DNA was purified, spotted onto a dot blot membrane and probed for both DNA-RNA hybrids (S9.6) and total DNA (methylene blue staining) (*Figure 6D*). The relative hybrid to DNA level was then normalised to a control experiment in which there was no gene in the ectopic expression cassette. Wild-type PcrA overexpression did not alter relative R-loop levels, whereas overexpression of the E224Q mutant led to a significant increase (~2.7 fold). This increase is comparable to the effect of deleting known R-loop suppression factors from bacterial cells such as *rnhC* or inhibiting *rho* using BCM in *E. coli* (*Raghunathan et al., 2018*) (see also *Figure 6G*).

With the aim of testing whether the interaction of PcrA with RNAP is important for R-loop regulation, we next overexpressed the CTD of WT PcrA or a CTD mutant (K727A) that does not interact with RNAP (*Sanders et al., 2017*). Based upon in vitro TEC supershifts (*Figure 1—figure supplement 1D*) and ex vivo pulldown assays (*Figure 6F* and *Figure 6—figure supplement 3A*), we knew that the free CTD inhibited the interaction between full length PcrA and RNAP. Therefore, we reasoned that overexpression of the CTD in cells should block endogenous wild-type PcrA from being recruited by RNAP. The relative R-loop levels in strains overexpressing the WT or mutant CTD were measured by dot-blot as before (*Figure 6G* and *Figure 6—figure supplement 3B–C*). Overexpression of the WT CTD caused a significant increase in R-loop levels compared to WT cells or the empty expression cassette control. This increase approached the levels associated with an *rnhC* deletion strain. However, replacing the wild-type CTD with a CTD mutant that fails to interact with RNAP reduced R-loop levels to a value not significantly different to the strain with an empty expression cassette. This supports the idea that the activity of PcrA suppresses R-loop formation and that this effect is dependent upon the ability of the CTD Tudor domain to interact with RNAP and/or other partner proteins. Finally, to determine whether this function of PcrA is conserved in other bacteria, we compared R-loop levels in WT and Δ*uvrD E. coli* strains (where deletion of *uvrD* is not lethal). We observed that loss of UvrD caused a significant increase in R-loops levels in the cell (*Figure 6—figure supplement 3D–E*).

## Discussion

The bacterial cell displays a remarkable ability to maintain genomic stability during rapid growth when many different DNA and RNA transactions occur simultaneously on the same chromosome. The UvrD-like family of helicases play a critical role at the interface of such pathways, orchestrating DNA repair as damage arises and helping to minimise conflicts that would otherwise occur between DNA replication and transcription (*Dillingham, 2011*). Several reports have identified physical or functional interactions between PcrA/UvrD and RNA polymerase which could underpin its emerging role in managing conflicts between transcription and replication or repair processes.

We showed here that PcrA binds tightly to a pre-formed TEC in vitro to form a complex that is stable during gel filtration, in contrast to complexes formed between free PcrA and RNAP. The complex was unaffected by removal of the upstream or downstream DNA duplex from the TEC, was still formed between the CTD (which lacks the helicase domains) and the TEC, and was not formed when

a truncated form of PcrA lacking the CTD was used. These observations suggest that the increased stability associated with the PcrA:TEC complex is not simply due to the helicase being able to engage with DNA/RNA that flanks the polymerase. Instead, the RNA polymerase might adopt a conformation in the TEC that favours the recruitment of PcrA via the CTD.

HDX-MS analysis confirmed that RNAP interacted with a face of the PcrA Tudor fold that is frequently observed to form protein interactions in other systems (*Musselman et al., 2012*; *Ruthenburg et al., 2007*). The site for this PcrA-CTD interaction on the RNA polymerase was identified as a short peptide motif in the β subunit that sits within the SI1 domain. This is the first protein-protein interaction reported for the enigmatic SI1 domain, which is a lineage-specific insert in the β subunit whose function remains mysterious (*Newing et al., 2020*). This interaction site is distinct from that bound by the TC-NER factor Mfd, which also engages with the beta subunit via a Tudor domain but at the β1 domain. In agreement with this and other data (*Kawale and Burmann, 2020*), we also found that PcrA and Mfd did not compete for RNAP in a physical interaction assay.

The short peptide motif to which PcrA binds is highly conserved within *Bacillus*-like SI1 domains but does not obviously align with β subunit sequences from more distantly related organisms. Indeed, because PcrA/UvrD is ubiquitous and its CTD so highly-conserved, we were initially surprised to find that the interaction with RNAP mapped to a highly variable domain. This paradox was resolved using bioinformatics, which revealed that the same motif is present widely in bacterial RNAPs, but that its positioning within the SI1 domain is different depending on the lineage. Relative to other more universal features in RpoB sequences, the interaction motif appears later in *B. subtilis*-like domains (which are representative of Firmicutes) than it does in *E. coli*-like domains (which are representative of Proteobacteria; see *Figure 2—figure supplement 2*). Note, however, that we did find examples of bacterial RpoB species for which we could find no clear helicase-interaction motif present in the SI1 domain. Our sequence searches proved rewarding as they revealed the presence of the same motif in other known PcrA-interacting proteins, including UvrB, YxaL, and YwhK. Moreover, this helicase interaction motif is also present in several poorly characterised proteins which might be previously unidentified interaction partners for PcrA. Where a structure is available, the motif always adopts a beta-hairpin with the highly conserved TGE triad protruding towards the solvent. The corollary of our observation that RNAP and UvrB employ the same structural element to recruit PcrA is that they might compete in the cell for the PcrA/UvrD helicase. This idea is supported directly by competition pulldown assays and is consistent with previous work in which the PcrA/UvrD CTD was shown to be important for interaction with both RNAP and UvrB (*Manelyte et al., 2009*; *Sanders et al., 2017*). However, the helicase interaction motif was absent from some known PcrA partners including the mismatch repair factor MutL. This is in agreement with recent work which showed that MutL binds and activates PcrA/UvrD via the 2B domain, rather than the CTD (*Ordabayev et al., 2018*; *Ordabayev et al., 2019*).

PcrA/UvrD, along with ppGpp, has been suggested to participate in an Mfd-independent transcription-coupled DNA repair pathway (*Epshtein et al., 2014*; *Pani and Nudler, 2017*). In this pathway, UvrD is thought to promote backtracking of RNAP stalled at bulky DNA lesions and then recruits NER factors including UvrB to repair the damage. We find here that RNAP and UvrB compete for PcrA interaction using the same conserved motif, meaning it is unclear how UvrB would be recruited to the UvrD:RNAP complex. Moreover, the role of UvrD in promoting strand-specific repair of DNA lesions remains controversial because a genome-wide analysis of cyclobutane pyrimidine dimers showed no evidence for Mfd-independent TC-NER and also that the levels of ppGpp do not change the repair rate in *E. coli* (*Adebali et al., 2017a*; *Adebali et al., 2017b*). Our data lead us to a new hypothesis to explain why PcrA/UvrD interacts with RNA polymerase.

The HDX-MS data with full-length PcrA showed that the core helicase domains interact with multiple elements surrounding the RNA and DNA exit channels, for instance the β-flap tip and the N-terminal part of the β′ subunit, as has been shown previously by crosslinking-MS data for the complex between *E. coli* RNAP and UvrD (*Epshtein et al., 2014*). Our HDX-MS data also identified an interaction with the SI1 domain which we can assign to the PcrA CTD. The exit channels for nucleic acids are distant from the SI1 domain, but the long linker between the helicase core and the CTD is sufficiently long to accommodate both interaction patches. Interestingly, the length of this linker varies in different PcrA orthologues in a manner that reflects the distance between the RNAP exit channels and SI1 domains in each system. The precise details of the PcrA-RNAP interaction will await a high-resolution structure of the complex, but it is possible to build a speculative physical model which

satisfies both our own HDX data and the crosslinking study of the *E. coli* system. (*Figure 7—figure supplement 1*).

The involvement of the main body of the helicase in RNAP interactions may also help to explain the phenotypes associated with CTD deletion. Given the apparently key role played by the PcrA/UvrD CTD as a protein interaction hub, it is surprising that its complete removal has little effect on nucleotide excision repair or replication-transcription conflicts in *E. coli* (*Merrikh et al., 2015*; *Sanders et al., 2017*). Similarly, in *B. subtilis* where PcrA is recruited to highly transcribed regions of the genome to facilitate replication through active transcription units, the CTD seems to be dispensable (*Merrikh et al., 2015*). A possible explanation lies in the fact that PcrA-RNAP association also involves the main body of the helicase including the core motor domain 2A, whose activity is largely unchanged by removal of the CTD. It is possible that residual weak interactions allow PcrA helicase core to function effectively in the absence of the CTD in vivo. However, it should also be acknowledged that stable interactions between the PcrA helicase core and RNAP cannot be detected between purified proteins (this and previous work *Gwynn et al., 2013*; *Sanders et al., 2017*), and consequently, we did not pursue HDX-MS experiments between these constructs. Therefore, at least in vitro, the presence of the CTD appears to facilitate binding of the core helicase domains near the RNA exit channels. Moreover, our observation of increased DNA-RNA hybrids in cells in which the free CTD is overexpressed provides a direct link between the RNAP binding function of the CTD and R-loop metabolism.

The location of the helicase core close to the RNA exit channel led us to consider roles for PcrA in manipulating the transcript. Due to the enzyme's inability to translocate on RNA or to unwind duplex RNAs, we hypothesised that PcrA might act to unwind DNA/RNA hybrids that are formed upstream of the transcription bubble. This idea was supported by experiments in *Bacillus subtilis* showing that R-loop concentration in cells was increased by two- to threefold by overexpressing dominant negative PcrA or free CTD (which blocks PcrA-RNAP interaction). This strongly suggests that an activity of wild-type PcrA, which is dependent upon interactions made by its CTD, is important for suppressing R-loops. Although out work does not establish the mechanism by which PcrA decreases R-loop levels in cells, we hypothesise that it does so by unwinding RNA-DNA hybrids that form behind the transcription elongation complex (*Figure 7*). The well-documented 3'>5' polarity of PcrA/UvrD helicase (*Bird et al., 1998*), together with its inability to move along RNA strands, suggests two possible working models for the unwinding of R-loops. In the first 'co-directional' model, the helicase translocates on the template strand and in the same direction as the RNA polymerase, unwinding R-loops as they form. Another possibility (the 'backtracking model') is that the helicase translocates on the non-template strand of the R-loop and in the opposite direction to transcription, indirectly unwinding the DNA/RNA hybrid by making the polymerase backtrack (*Epshtein et al., 2014*). In this scenario the lid domain of RNAP, rather than the helicase itself, would act as the 'ploughshare' to separate the two strands. We favour the first model because our HDX protection data places the 2A domain of PcrA (which is at the front of the moving helicase) in direct contact with the RNA exit region, suggesting that the helicase translocates towards the RNA polymerase. Indeed, the close contact between PcrA and RNAP suggested by our data could mean that R-loops are unwound as they are formed at the exit channels, rather than being targeted after they have extended behind the TEC. However, we also note that previous observations of UvrD-dependent backtracking of RNAP are more consistent with the second backtracking model (*Epshtein et al., 2014*; *Sanders et al., 2017*). In these models, we propose that the role of the CTD is simply to target the helicase activity to its physiological substrate in vivo (i.e. to increase the local concentration of PcrA near R-loops), rather than to catalytically activate the DNA motor protein in response to its engagement with RNAP. This view is consistent with our observation that free CTD can block R-loop suppression through dominant negative effects on wild-type PcrA, and also with previously published work in which we have observed only moderate effects of adding RNAP and/or removing the CTD on the ATPase and helicase activities of PcrA (*Gwynn et al., 2013*; *Velankar et al., 1999*).

Further experiments are now required to test the idea that PcrA unwinds co-transcriptional R-loops more directly, but the model is appealing for several reasons. A role for PcrA in suppressing R-loops can explain its function in alleviating replication:transcription conflicts as well as why the helicase is enriched at highly transcribed rRNA and tRNA genes that have been shown to be prone to R-loop formation in yeast and human cells (*Boubakri et al., 2010*; *Chan et al., 2014*; *Chen et al., 2017*; *Guy et al., 2009*; *Merrikh et al., 2015*). Indeed, recent work in *E. coli* has shown that *uvrD* is

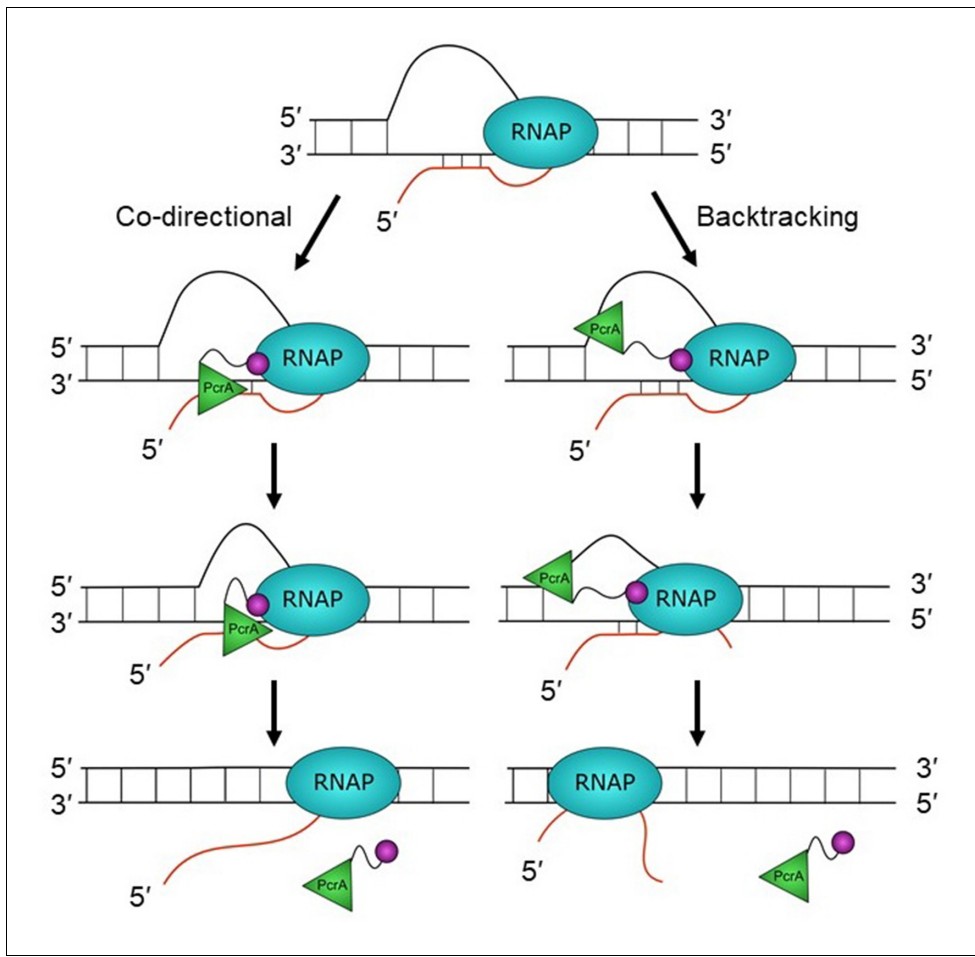

**Figure 7.** Hypothetical models for PcrA-dependent R-loop suppression during transcription. R-loops form during transcription but lead to genomic instability and are therefore targeted for removal by PcrA. In the co-directional model, PcrA helicase (green) interacts with an R-loop-associated RNAP (blue) via the PcrA-CTD (magenta). It then engages the template DNA strand before translocating in the 3′>5′ direction and directly unwinding the DNA: RNA hybrid. In this model, R-loops may be unwound as they are formed if the helicase is in close contact with the TEC. In the backtracking model, PcrA binds to the displaced (non-template) strand behind RNAP and pulls it backwards, thereby unwinding the R-loop indirectly (see main text for further details and Discussion).

The online version of this article includes the following figure supplement(s) for figure 7:

**Figure supplement 1.** Working model for the interaction between PcrA and a TEC.

synthetically lethal with *rnhA*, in agreement with a role for this helicase in unwinding R-loops that may be distinctive from those dealt with by other R-loop suppression factors (*Wolak et al., 2020*). An R-loop suppression function might also explain why loss of PcrA leads to hyper-recombination and the formation of RecFOR- and RecA-dependent toxic recombination intermediates (*Arthur and Lloyd, 1980*; *Moreno-Del Alamo et al., 2020*; *Petit and Ehrlich, 2002*; *Veaute et al., 2005*). The displaced ssDNA within an unresolved R-loop may act as a 'frustrated' substrate for ssDNA gap repair, resulting in RecA recruitment and strand exchange but failing to be processed correctly into repaired products because of the presence of the R-loop and TEC. Finally, a very recent study involving depletion of PcrA in *rnhB*, *rnhC*, and *dinG* backgrounds led to the conclusion that PcrA alleviates replication-transcription conflicts, with biochemical analysis supporting the idea (broadly similar to that presented here) that the helicase activity of PcrA acts to unwind and remove R-loops in concert with R-loop nucleases (*Moreno-Del Álamo et al., 2021*).

## Materials and methods

### Strain construction

All strains used in this work are listed in *Table 3*. The plasmids and primers used to create them are described in *Tables 4* and *5*, respectively. Standard techniques were used for strain construction (*Harwood and Cutting, 1990*). *B. subtilis* was transformed using the LM-MD method or an optimised two-step starvation procedure (*Anagnostopoulos and Spizizen, 1961*; *Burby and Simmons, 2017*) Integrations within the ectopic locus were verified by PCR and/or sequencing.

### Protein expression and purification

His-tagged *Bacillus subtilis* RNA polymerase was purified as described previously from the MH5636 strain (*Gwynn et al., 2013*; *Qi and Hulett, 1998*). Wild-type and mutant *B. subtilis* PcrA and biotinylated PcrA were purified as described (*Gwynn et al., 2013*). Plasmids expressing PcrA ΔCTD (residues 1–653) and PcrA V448C were generated by site-directed mutagenesis of the wild-type construct using the Quikchange II XL kit (Agilent Technologies). In the last gel filtration chromatography of PcrA V448C, DTT was omitted to allow efficient fluorescent labelling. Fluorescent PcrA was prepared by mixing the V448C mutant with Cy3-maleimide (Cytiva) in a 1:2 molar ratio and incubating overnight at 4°C. The reaction was quenched by the addition of 5 mM DTT. The free dye was separated from the labelled protein by loading the mixture on a Superdex 200 5/150 GL. Fluorescent labelling efficiency was calculated spectrophotometrically following the manufacturer's instructions. A plasmid expressing the PcrA-CTD was generated by PCR amplification from the wild-type plasmid followed by sub-cloning into the pET47b plasmid. *B. subtilis* PcrA-CTD was then purified in

**Table 3.** List of *B. subtilis* and *E. coli* strains used in this work.

| Strain | Genotype | Reference | Plasmid to generate the strain | Parent strain |
|---|---|---|---|---|
| MH5636 | rpoC-His10::cat trpC2 | *Qi and Hulett, 1998* | - | JH642 |
| 1a1 | trpC2 | *Koo et al., 2017* | - | - |
| IU79 | lacA::Pxyl-myc-PcrA::mls ins[b] pMAP39 in pcrA | This work | pMAP39 (*Petit et al., 1998*) and pBS2EXylRP$_{xylA}$ $_{(V2)}$- myc-PcrA | 1a1 |
| IU6 | amyE::Phyperspank::spec HO trpC2 | This work | pDRIII (HO) | 1a1 |
| IU35 | amyE::Phyperspank::spec CD trpC2 | This work | pDRIII (CD) | 1a1 |
| IU41 | amyE::Phyperspank-mycPcrA::spec CD trpC2 | This work | pDRIII(CD)-mycPcrA | 1a1 |
| IU56 | amyE::Phyperspank-mycPcrA-E224Q::spec CD trpC2 | This work | pDRIII(CD)-mycPcrA-E224Q | 1a1 |
| BKK00550 | Δmfd:kan trpC2 | *Koo et al., 2017* | - | 1a1 |
| IU60 | amyE::Phyperspank::spec Δmfd:kan trpC2 | This work | pDRIII (CD) | BKK00550 |
| IU61 | amyE::Phyperspank-myc-PcrA::spec Δmfd:kan trpC2 | This work | pDRIII(CD)-mycPcrA | BKK00550 |
| IU62 | amyE::Phyperspank-myc-PcrA-E224Q::spec Δmfd:kan trpC2 | This work | pDRIII(CD)-mycPcrA-E224Q | BKK00550 |
| IU65 | amyE::Phyperspank-myc-PcrAΔCTD-E224Q::spec trpC2 | This work | pDRIII(CD)-mycPcrAΔCTD-E224Q | 1a1 |
| IU66 | amyE::Phyperspank-myc-PcrAΔCTD-E224Q::spec Δmfd:kan trpC2 | This work | pDRIII(CD)-mycPcrAΔCTD-E224Q | BKK00550 |
| BKK28620 | trpC2 ΔrnhC::kan | *Koo et al., 2017* | - | 1a1 |
| IU3 | amyE::Phyperspank::spec(HO) trpC2 | This work | pDRIII(HO) | BKK28620 |
| IU5 | amyE::Phyperspank-myc-CTD::spec (HO) trpC2 | This work | pDRIII(HO)-mycCTD | BKK28620 |
| IU9 | amyE::Phyperspank-myc-CTD-K727A::spec (HO) trpC2 | This work | pDRIII(HO)-mycCTD-K727A | BKK28620 |
| TB28 | *E. coli* ΔlacIZYA | *Bernhardt and de Boer, 2004* | - | MG1655 |
| N6632 | *E. coli* ΔuvrD::dhfr | *Guy et al., 2009* | - | MG1655 |

**Table 4.** Plasmids used in this work.

| Plasmid name | Vector | Insert(s) | Reference |
|---|---|---|---|
| pET22b-PcrA | pET22b | PcrA | *Gwynn et al., 2013* |
| pET22b-bioPcrA | pET22b | BioPcrA | *Gwynn et al., 2013* |
| pET22b-PcrAV448C | pET22b | PcrA V448C | This work |
| pET22b-PcrAΔCTD | pET22b | PcrAΔCTD | This work |
| pET47b-CTD | pET47b | CTD | This work |
| pET28a-rpoB | pET28a | RNAP β subunit | This work |
| pET28a-rpoB-E310A | pET28a | RNAP β subunit E310A | This work |
| pET28a-rpoB-E310K | pET28a | RNAP β subunit E310K | This work |
| pET47b-UvrB | pET47b | UvrB | This work |
| pET47b-UvrB-E233A | pET47b | UvrB E233A | This work |
| pET22b-Mfd | pET22b | Mfd | This work |
| pBS2EXyl-mycPcrA | pBS2EXylRPxylA (V2) pBS0EPliaI | mycPcrA | *Popp et al., 2017* |
| pDRIII(HO) | pDRIII(HO) | - | *Fisher et al., 2017* |
| pDRIII(CD) | pDRIII(CD) | - | This work |
| pDRIII(CD)-mycPcrA | pDRIII(CD) | mycPcrA | This work |
| pDRIII(HO)-mycPcrA | pDRIII(HO) | mycPcrA | This work |
| pDRIII(CD)-mycPcrA-E224Q | pDRIII(CD) | PcrA-E224Q | This work |
| pDRIII(CD)-mycPcrAΔCTD-E224Q | pDRIII(CD) | PcrAΔCTD-E224Q | This work |
| pDRIII(HO)-mycCTD | pDRIII(HO) | mycCTD | This work |
| pDRIII(HO)-mycCTD-K727A | pDRIII(HO) | mycCTD-K727A | This work |
| pMAP39 | pBSspec+ | (nt 201 to 699 of pcrA) at HincII | *Petit et al., 1998* |

the same way as *Geobacillus stearothermophilus* PcrA-CTD but the gel-filtration chromatography was substituted for MonoS (Cytiva) ion-exchange chromatography (*Sanders et al., 2017*).

*B. subtilis uvrB*, *mfd*, and *rpoB* were amplified by colony PCR from *B. subtilis 168* single colonies and cloned into pET47b, pET22b, and pET28a, respectively (cloning oligonucleotides available in the Supplementary Methods and *Table 5*). Mutations to generate plasmids for expression of UvrB E233A, RpoB E310A, and RpoB E310K were made by site-directed mutagenesis of the wild type constructs using the Quikchcange II XL kit (Agilent). *B. subtilis* Mfd was purified in the same way as *E. coli* Mfd, as described previously (*Chambers et al., 2003*). His-tagged *B. subtillis* UvrB was expressed in BL21 (DE3) cells. Following induction at mid-log phase with 1 mM IPTG, cells were grown overnight at 18°C. Cell pellets were resuspended in 20 mM Tris–HCl pH 7.9, 300 mM NaCl, 20 mM imidazole, 0.1 mM DTT supplemented with EDTA-free cOmplete protease inhibitor cocktail (Roche) and lysed by sonication. The soluble fraction was loaded on a 5 ml HisTrap column (Cytiva) in the same buffer and eluted with an imidazole gradient. A portion of the material was supplemented with 3C protease and the two separate samples were then dialysed against 20 mM Tris-HCl pH 7.9, 300 mM NaCl, 20 mM imidazole and 0.1 mM DTT overnight at 4°C. For the sample that had been cleaved with 3C, the protein was loaded again onto a HisTrap column to remove the cleaved tag and the protease (which is itself his-tagged). The flow through was diluted into 20 mM Tris-HCl, 1 mM DTT to reduce the salt concentration before loading onto a MonoQ column (Cytiva) equilibrated in 20 mM Tris-HCl, 100 mM NaCl, 1 mM DTT. The protein was eluted with a NaCl gradient and fractions containing UvrB were pooled and dialysed against storage buffer (20 mM Tris-HCl pH 7.5, 200 mM NaCl, 2 mM EDTA, 1 mM DTT and 10% glycerol) overnight at 4°C. The uncleaved sample was purified in the same way but without the second HisTrap chromatography step.

His-tagged *B. subtilis* RNAP β subunit was expressed and purified in the same manner as UvrB. The uncleaved sample was then further purified using a heparin column equilibrated in 20 mM Tris-HCl, 100 mM NaCl, 1 mM DTT followed by elution with a NaCl gradient. The protein was dialysed

**Table 5.** Oligonucleotides used in this work for cloning (all sequences written 5′>3′).

| Name | Sequence | Purpose |
|---|---|---|
| PcrA_V448C_F | AAGCGATTCAGCAGTGTGATTTTATCG | SDM |
| PcrA_V448C_R | CGATAAAATCACACTGCTGAATCGCTT | SDM |
| His-PcrA_CTD_XmaI_F | GACTCCCGGGAAAGAAACAAGAGCGACGTC | PcrA CTD (664-739) subcloning in pET47b |
| BsuPcrA_BamHI_R | GATCGGATCCTTACTGCTTTTCAATAGGAGCAAATG | PcrA CTD (664-739) subcloning in pET47b |
| PcrA_E224Q_F | CATCCACGTTGATCAGTATCAGGATACGAAC | SDM |
| PcrA_E224Q_R | GTTCGTATCCTGATACTGATCAACGTGGATG | SDM |
| BSuPcrA_deltaCTD_F | CCTAAATGAGAAATAAGAAACAAG | SDM |
| BSuPcrA_deltaCTD_R | CTTGTTTCTTATTTCTCATTTAGG | SDM |
| bsurpob_ndeI_F | CAGGTGCATATGTTGACAGGTCAACTAGTTCAGTATG | RpoB subcloning in pET28a |
| bsurpob_xhoI_R | TTAATTCTCGAGTTATTCTTTTGTTACTACATCGCGTTC | RpoB subcloning in pET28a |
| rpoB_E301A_F | GATCCTGAAACAGGAGCAATCCTTGCTGAAAAAG | SDM |
| rpoB_E301A_R | CTTTTTCAGCAAGGATTGCTCCTGTTTCAGGATC | SDM |
| rpoB_E310K_F | GATCCTGAAACAGGAAAAATCCTTGCTGAAAAAG | SDM |
| rpoB_E310K_R | CTTTTTCAGCAAGGATTTTTCCTGTTTCAGGATC | SDM |
| BSuUvrB_SmaI_F | AGCAGCCCGGGGTGAAAGATCGCTT TGAGTTAGTCTCGAAATATC | UvrB subcloning in pET47b |
| BsuUvrB_XhoI_R | GCATCTCGAGTCATCCTTCCGCT TTTAGCTCTAAAAGTAAATC | UvrB subcloning in pET47b |
| BsuUvrB_E233A_F | GCTGACAGGAGCAATTCTCGGCGAC | SDM |
| BsuUvrB_E233A_R | GTCGCCGAGAATTGCTCCTGTCAGC | SDM |
| bMfd_5′NdeI_F | TTAATCATATGGACAACATTCAAACCTTT | Mfd subcloning in pET22b |
| bMfd_XhoI_3′_R | ATTAACTCGAGTTACGTTGATGAAATGGTTTG | Mfd subcloning in pET22b |
| bMfd_mutA2586G_F | CCTGACGCGAAGGTAGCGTATGCGCATGGGAAAATG | SDM |
| bMfd_mutA2586G_R | CATTTTCCCATGCGCATACGCTACCTTCGCGTCAGG | SDM |
| bMfdmutT2361C_F | CGCGTACGCTGCACATGTCTATGCTTG | SDM |
| bMfd_mutT2361C_R | CAAGCATAGACATGTGCAGCGTACGCG | SDM |
| pDRIII-inver-upsR-BlpI | TACTTAGCTAAGCCTAACTC ACATTAATTGCGTTGCG | Invert MCS in pDRIII |
| pDRIII-inver-downsF-BamHI | TAATTTGGATCCCTAAGCAGAAGGCCATCCTG | Invert MCS in pDRIII |
| CTD_HindIII_F | ATCGTAAGCTTAAAGAAACAAGAGCGACGTC | CTD subcloning in pDRIII |
| CTD_SphI_R | GCTTTGCATGCTTACTGCTTTTCAATAGGAGCAAATG | CTD and PcrA subcloning in pDRIII |
| myc-PcrA_5′SalI_RBS_F | CGTTGTCGACAGGAGGTATACATATGGAGCAAAG | PcrA subcloning in pDRIII |
| PcrA_K727A_F | CTGTCGGCGTGGCACGCCTGTTAGCAG | SDM |
| PcrA_K727A_R | CTGCTAACAGGCGTGCCACGCCGACAG | SDM |

against 20 mM Tris-HCl pH 7.9, 300 mM NaCl, 2 mM EDTA, 1 mM DTT, and 10% glycerol overnight at 4°C.

## Analytical size-exclusion chromatography

Transcription elongation complexes (TEC) were formed as described previously (*Sidorenkov et al., 1998*) using scaffold 1 (*Table 6*) and the RNA sequence: AUCGAGAGG (IBA life sciences). Briefly, the RNA oligonucleotide was incubated with the template strand (TS) for 5 min at 45°C in buffer T (50 mM Tris pH 7.9, 150 mM NaCl, 10 MgCl$_2$ and 1 mM DTT) and cooled to room temperature at a rate of 1°C/min. Then, RNAP was added and the solution was incubated at room temperature for 10 min. Finally, non-template strand (NTS) was added and incubated at 37°C for 10 min. The final concentration of each component was: 118 nM TS, 236 nM RNA, 2.36 µM NTS, and 66 nM RNAP. To form the PcrA-TEC complex, the TEC was concentrated 10-fold using Amicon Ultra-0.5 3 kDa

**Table 6.** Oligonucleotides used in this work for assembling the TEC (all sequences 5'>3').

| Name | Strand | Modification | Sequence (5'−3') |
|---|---|---|---|
| RNA I | - | - | AUCGAGAGG |
| Standard scaffold | TS | 5' Cy5 | TGTCACTTCGCCGTGTCCCTCTCGATGGCTGTAAG TATACT |
| | NTS | | AGTATACTTACAGCCATCGAGAGGGACACGGCGAAGTG ACA |
| Downstream gap in TS | TS | 5' Cy5 | CGTGTCCCTCTCGATGGCTGTAAGTATACT |
| | NTS | | AGTATACTTACAGCCATCGAGAGGGACACGGCGAAGTGACA |
| Upstream gap in NTS | TS | 5' Cy5 | TGTCACTTCGCCGTGTCCCTCTCGATGGCTGTAAGTATAC |
| | NTS | | AGCCATCGAGAGGGACACGGCGAAGTGACA |
| Short scaffold | TS | 5' Cy5 | CGTGTCCCTCTCGATGGCT |
| | NTS | | AGCCATCGAGAGGGACACG |
| No duplex downstream | TS | 5' Cy5 | CGTGTCCCTCTCGATGGCTGTAAGTATACT |
| | NTS | | AGTATACTTACAGCCATCGAGAGGGACACG |
| No duplex upstream | TS | 5' Cy5 | TGTCACTTCGCCGTGTCCCTCTCGATGGCT |
| | NTS | | AGCCATCGAGAGGGACACGGCGAAGTGACA |

NMWL centrifugal filters and then incubated with 4.6 µM PcrA for 10 min at room temperature. Samples were loaded onto a Superdex 200 Increase 5/150 GL (Cytiva) connected to an AKTA FPLC instrument (Cytiva) in buffer T (50 mM Tris pH 7.9, 150 mM NaCl, 10 MgCl$_2$ and 1 mM DTT) at room temperature. Samples were directly injected in the injection loop prior to the chromatography. Where necessary, samples were collected to analyse their content by SDS-PAGE.

## Electrophoretic mobility shift assays

TECs were assembled as above but using different DNA:RNA scaffolds containing fluorescently labelled oligonucleotides (Table 6), the buffer supplemented with 2.5% (v/v) Ficoll and analysed for their ability to bind to PcrA. Wild type or fluorescent PcrA at the concentration indicated was added to the TEC at room temperature for 10 min before loading the samples onto a 4.5% polyacrylamide TBE-PAGE gel. The gels were imaged using a Typhoon FLA 9500 (Cytiva).

## Analysis of protein:protein interactions using pulldown assays

*B. subtilis* cell extracts were prepared for ex vivo pulldowns as described previously (Gwynn et al., 2013) using strains 1A1 or IU79, the latter supplemented with 1% xylose to induce the expression of myc-tagged PcrA. 1 µM biotin-tagged proteins were used as baits on streptavidin beads, whereas 2 µM his-tagged proteins were used with Ni-NTA beads. The wash and dilution buffer (20 mM Tris pH 7.5, 150 mM NaCl, 0.1% Triton X-100) was supplemented with1 mM EDTA for experiments with the streptavidin beads and 20 mM imidazole for Ni-NTA beads. For the in vitro pulldown, purified β subunit was used at 2 µM concentration. For experiments that contained competitors 1.5 µM PcrA or PcrA-CTD, 1.5 µM Mfd and 3 µM UvrB were used. In vitro pulldowns were analysed by Coomassie staining the SDS-PAGE gel. Ex vivo pulldowns were analysed by western blotting using a monoclonal anti-RNAP β subunit antibody (Abcam; ab202891) or a monoclonal anti-myc antibody (Proteintech; 67447–1-Ig). In the latter experiments, a portion of the gel was excised before transfer to the blotting membrane to confirm uniform bait levels by Coomassie staining, except where this was not possible with UvrB due to its similar molecular weight to PcrA. Images were quantified using ImageQuant (Cytiva).

## Hydrogen-deuterium exchange coupled to mass spectrometry

HDX-MS experiments were performed using an automated robot (LEAP Technologies) coupled to an M-Class Acquity UPLC, HDX manager (Waters Ltd.), and Synapt G2S-i mass spectrometer. For the CTD-RNAP experiments, protein samples were prepared containing either or both 5 µM RNAP and 10 µM CTD. For the PcrA-RNAP experiments, samples were prepared using 0.5 µM RNAP and/or 2 µM PcrA. Thirty µl of the protein solution in sample buffer (10 mM potassium phosphate buffer

pH 7.4) was added to 135 μl deuterated buffer (10 mM potassium phosphate buffer in deuterated water, pD 7.4) and incubated at 4℃ for 0.5, 2, 5, and 10 min. After labelling, the reaction was terminated by the addition of 50 μl of sample to 100 μl quench buffer (10 mM potassium phosphate, 2 M guanidine-HCl, pH 2.2) resulting in a final pH of approximately 2.5. Fifty μl of quenched sample were loaded onto an immobilised ethylene-bridged hybrid (BEH) pepsin column (Waters Ltd.). The resulting peptides were passed through a VanGuard BEH C18 pre-column and a C18 column (Waters Ltd.) and separated by gradient elution of 0–40% MeCN (0.1% v/v formic acid) in $H_2O$ (0.3% v/v formic acid). For the CTD-RNAP experiments, quench and pepsin wash buffers were supplemented with 0.05% (w/v) DDM.

The UPLC was interfaced to the mass spectrometer via electrospray ionisation. HDMSE and dynamic range extension modes (Data Independent Analysis couple with IMS separation) were used to separate peptides by IMS prior to fragmentation in the transfer cell. Data were analysed using PLGS and DynamX software provided with the mass spectrometer. The restrictions applied for peptide identification in DynamX were the following: minimum intensity 1000, minimum products per amino acid 0.3, maximum sequence length 25, maximum ppm error 6, replication file threshold 4. The difference plots were generated using the in-house developed algorithm Paved (*Cornwell et al., 2018*), Deuteros (*Lau et al., 2019*) and the data represented in Graphpad Prism 7.0. Difference plots were then mapped onto structures in Chimera (*Pettersen et al., 2004*) and PyMOL using available structures and/or homology models generated by Swiss-model (*Waterhouse et al., 2018*). Four technical repeats were performed for each HDX-MS experiment, each using single protein samples pooled from independent preparations. The data have been deposited with the PRIDE database (Accession number PXD025332).

## Bioinformatic analysis

Protein orthologues were searched for in the reference proteomes of the Protein Information Resource using the *B. subtilis* protein sequence as a query (*Chen et al., 2011*). Obtained sequences were manually curated to remove sequences that were substantially smaller than the *B. subtilis* sequence or that contained large gaps. The conservation logo was created using WebLogo (*Crooks et al., 2004*) and sequence alignment figure created with ESPript 3.0 (*Robert and Gouet, 2014*). *B. subtilis* proteins' homology models were created using SWISS-MODEL with the indicated PDB files (*Waterhouse et al., 2018*). Protein-protein interaction docking analysis was performed using Haddock 2.2 and the best structures from the clusters with the lowest HADDOCK score were compared (*van Zundert et al., 2016*).

## CD spectroscopy

CD spectra were collected using a JASCO J-810 spectropolarimeter fitted with a Peltier temperature control (Jasco UK). A total of 0.5 mg/ml protein samples were buffer exchanged into phosphate buffered saline (PBS; 8.2 mM $NaH_2PO_4$, 1.8 mM $KH2PO_4$, 137 mM NaCl and 2.7 mM KCl (pH 7.4)) by 16 hr dialysis at 4℃ using a membrane with a MWCO of 10 kDa. Using 0.25 mg/ml of protein in a 0.1 cm quartz cuvette, data were acquired across a 190–206 nm absorbance scan with a scan rate of 100 nm/min at 20℃. Raw data was normalised to molar ellipticity (MRE (deg.$cm^2$.$dmol^{-1}$)) using calculation of the concentration of peptide bonds and the cell path length. A buffer only baseline was subtracted from all datasets.

## Helicase assays

Helicase assays were carried out essentially as described previously with the following changes (*Gwynn et al., 2013*). The reactions were performed at 20℃ in 50 mM Tris pH 7.5, 50 mM NaCl, 2 mM $MgCl_2$, and 1 mM DTT and started by the addition of the protein. The list of DNA and RNA substrates used is shown in *Table 7*. The products were run on 12% acrylamide TBE gels, imaged using a Typhoon phosphorimager and quantified using ImageQuant software.

## ATPase assay

The ATPase activities of PcrA and UvrB were measured using an enzyme linked assay in which ATP hydrolysis is coupled to NADH oxidation (*Kiianitsa et al., 2003*). The assay was carried out at 25℃ using 1 nM PcrA, 50 nM UvrB, and 5 μM nucleic acids (as indicated). For measurements of PcrA

**Table 7.** Oligonucleotides used in this work for helicase assays (all sequences 5′>3′).

| Oligo name | Type | Sequence (5′−3′) |
|---|---|---|
| IU_1 | DNA | GGGAGCCGGTCTGCGTCTGGTGTACTCTTCTGCTTTCTCG |
| IU_1R | RNA | GGGAGCCGGUCUGCGUCUGGUGUACUCUUCUGCUUUCUCG |
| IU_2 | DNA | CCAGACGCAGACCGGCTCCC |
| IU_2R | RNA | CCAGACGCAGACCGGCUCCC |
| IU_3 | DNA | GGGAGCCGGTCTGCGTCTGG |
| IU_3R | RNA | GGGAGCCGGUCUGCGUCUGG |

ATPase activity the reaction buffer contained 50 mM Tris pH 7.5, 50 mM NaCl, 5 mM MgCl$_2$, and 5 mM DTT. The method used for UvrB was as described previously (*Webster et al., 2012*).

## Growth curves

Single colonies were grown overnight in LB at 30°C. Each strain was diluted to an OD$_{600}$ of 0.01 in LB supplemented with 1 mM IPTG and then grown at 37°C. Absorbance measurements were taken at the indicated times, averaged and plotted using Graphpad Prism.

## RNA/DNA hybrid dot blot

Single colonies were grown overnight in LB at 30°C. Cultures were diluted to an OD$_{600}$ of 0.05 into fresh LB medium supplemented with 1 mM IPTG for both *B. subtilis* morning and overnight inductions. For the FL-PcrA overexpression experiment, overnight cultures were induced the next morning at the time that they were diluted. For the CTD experiments, induction was started while inoculating the single colonies for the overnight growth. Cultures were grown until an OD$_{600}$1.1–1.2 and genomic DNA was purified using the DNeasy Blood and Tissue (Qiagen) or the GenElute Bacterial Genomic DNA Kit (Sigma-Aldrich) following manufacturer's instructions. gDNA was quantified by Nanodrop and a fraction of DNA was treated with 5 U RNase H (NEB) for 1 hr at 37°C. gDNA serial dilutions and the RNase treated gDNA were spotted on a positively charged Hybond-N+ nylon membrane (Amersham) using a dot-blot apparatus. The DNA was probed with the S9.6 antibody (1:1000 dilution, Millipore) in 1% BSA/TBST overnight at 4°C after UV-crosslinking (0.12 J/cm$^2$) and blocking the membrane with 5% milk/TBST for 1 hr at RT. An anti-mouse HRP antibody (1:10000 dilution, Santa Cruz Biotechnology) was used as secondary antibody. Images were acquired with Odyssey Fc (Li-COR Biosciences). DNA loading was calculated by staining with 0.05% methylene blue in 0.5 M sodium acetate buffer (pH 5.2) after washing the membrane with 5% acetic acid as described previously (*Ko et al., 2010*; *Raghunathan et al., 2018*). Images were quantified using ImageQuant (Cytiva). Error bars show the SEM of at least three independent experiments.

## Immunoblotting

*B. subtilis* cells were grown until stationary phase in LB at 37°C, harvested by centrifugation and the pellet was resuspended in SSC prior to sonication. OD$_{600}$ was used to normalise the loading of samples in the SDS-PAGE gel. The gel was cut into two parts prior to transfer, and the upper portion was Coomassie stained to use as loading control and the lower portion was transferred to a PVDF membrane by electroblotting. Myc-tagged PcrA was detected using a monoclonal anti-myc antibody (Proteintech) followed by an anti-mouse-HRP goat antibody (Santa Cruz Biotechnology).

## Mfd Pulldown-Tandem mass tagging mass spectrometry (TMT-MS)

*B. subtilis* lysates were prepared and pulldown performed using the 1A1 strain as described above. Two conditions were tested: 1 μM biotinylated Mfd and no-bait control. Samples were then analysed by TMT-MS to determine which prey proteins were enriched in the pulldown compared to control.

Pulled-down samples were reduced (10 mM TCEP, 55°C for 1 hr), alkylated (18.75 mM iodoacetamide, room temperature for 30 min.) and then digested from the beads with trypsin (2.5 μg trypsin; 37°C, overnight). The resulting peptides were then labelled with TMT ten-plex reagents according to the manufacturer's protocol (Thermo Fisher Scientific, Loughborough, LE11 5RG, UK), and the labelled samples pooled and desalted using a SepPak cartridge according to the manufacturer's

instructions (Waters, Milford, Massachusetts, USA). Eluate from the SepPak cartridge was evaporated to dryness and resuspended in buffer A (20 mM ammonium hydroxide, pH 10) prior to fractionation by high-pH reversed-phase chromatography using an Ultimate 3000 liquid chromatography system (Thermo Scientific). In brief, the sample was loaded onto an XBridge BEH C18 Column (130 Å, 3.5 µm, 2.1 mm X 150 mm, Waters, UK) in buffer A and peptides eluted with an increasing gradient of buffer B (20 mM Ammonium Hydroxide in acetonitrile, pH 10) from 0 to 95% over 60 min. The resulting fractions (four in total) were evaporated to dryness and resuspended in 1% formic acid prior to analysis by nano-LC MSMS using an Orbitrap Fusion Tribrid mass spectrometer (Thermo Scientific).

High-pH RP fractions were further fractionated using an Ultimate 3000 nano-LC system in line with an Orbitrap Fusion Tribrid mass spectrometer (Thermo Scientific). In brief, peptides in 1% (vol/vol) formic acid were injected onto an Acclaim PepMap C18 nano-trap column (Thermo Scientific). After washing with 0.5% (vol/vol) acetonitrile 0.1% (vol/vol) formic acid, peptides were resolved on a 250 mm × 75 µm Acclaim PepMap C18 reverse phase analytical column (Thermo Scientific) over a 150 min organic gradient with a flow rate of 300 nl min$^{-1}$. Solvent A was 0.1% formic acid and Solvent B was aqueous 80% acetonitrile in 0.1% formic acid. Peptides were ionised by nano-electrospray ionisation at 2.0kV using a stainless-steel emitter with an internal diameter of 30 µm (Thermo Scientific) and a capillary temperature of 275°C.

All spectra were acquired using an Orbitrap Fusion Tribrid mass spectrometer controlled by Xcalibur 2.1 software (Thermo Scientific) and operated in data-dependent acquisition mode using an SPS-MS3 workflow. FTMS1 spectra were collected at a resolution of 120 000, with an automatic gain control (AGC) target of 200 000 and a max injection time of 50 ms. Precursors were filtered with an intensity threshold of 5000, according to charge state (to include charge states 2–7) and with monoisotopic peak determination set to peptide. Previously interrogated precursors were excluded using a dynamic window (60 s +/−10ppm). The MS2 precursors were isolated with a quadrupole isolation window of 1.2 m/z. ITMS2 spectra were collected with an AGC target of 10,000, max injection time of 70 ms and CID collision energy of 35%.

For FTMS3 analysis, the Orbitrap was operated at 50 000 resolution with an AGC target of 50 000 and a max injection time of 105 ms. Precursors were fragmented by high energy collision dissociation (HCD) at a normalised collision energy of 60% to ensure maximal TMT reporter ion yield. Synchronous Precursor Selection (SPS) was enabled to include up to 5 MS2 fragment ions in the FTMS3 scan.

The raw data files were processed and quantified using Proteome Discoverer software v2.1 (Thermo Scientific) and searched against the UniProt *Bacillus subtilis* (strain 168) database (downloaded December 2018: 4284 entries) using the SEQUEST HT algorithm. Peptide precursor mass tolerance was set at 10ppm, and MS/MS tolerance was set at 0.6 Da. Search criteria included oxidation of methionine (+15.995 Da), acetylation of the protein N-terminus (+42.011 Da) and Methionine loss plus acetylation of the protein N-terminus (−89.03 Da) as variable modifications and carbamidomethylation of cysteine (+57.021 Da) and the addition of the TMT mass tag (+229.163 Da) to peptide N-termini and lysine as fixed modifications. Searches were performed with full tryptic digestion and a maximum of two missed cleavages were allowed. The reverse database search option was enabled and all data was filtered to satisfy a false discovery rate (FDR) of 5%. Only the proteins with a high FDR confidence and more than one unique peptide were accepted as hits. Fold change was calculated using the beads-only control.

## Acknowledgements

This work was funded by the European Union via the DNAREPAIRMAN innovative training network (Urrutia-Irazabal, Savery and Dillingham) and by a BBSRC ALERT grant (Ault and Sobott; BB/M012573/1). We are grateful to Abigail Smith and Gwendolyn Brouwer for helpful discussions, to Harry Thompson for technical assistance with CD measurements, to Kate Heesom and Phil Lewis for technical assistance with TMT-MS, to Peter McGlynn, Heath Murray, Marie-Agnès Petit, Houra Merrikh and Christopher Merrikh for advice and the sharing of plasmids and strains used in this work, and to Peter Lewis for generously sharing the co-ordinates of the *B. subtilis* transcription elongation complex ahead of public release.

## Additional information

### Funding

| Funder | Grant reference number | Author |
|---|---|---|
| H2020 European Research Council | DNAREPAIRMAN Grant agreement ID: 722433 | Inigo Urrutia-Irazabal<br>Nigel J Savery<br>Mark Simon Dillingham |
| Biotechnology and Biological Sciences Research Council | BB/M012573/1 | James R Ault<br>Frank Sobott |

The funders had no role in study design, data collection and interpretation, or the decision to submit the work for publication.

### Author contributions

Inigo Urrutia-Irazabal, Conceptualization, Data curation, Formal analysis, Investigation, Methodology, Writing - original draft, Writing - review and editing; James R Ault, Data curation, Formal analysis, Supervision, Methodology; Frank Sobott, Supervision, Funding acquisition, Writing - review and editing; Nigel J Savery, Conceptualization, Supervision, Funding acquisition, Writing - review and editing; Mark S Dillingham, Conceptualization, Supervision, Funding acquisition, Writing - original draft, Writing - review and editing

### Author ORCIDs

Inigo Urrutia-Irazabal http://orcid.org/0000-0003-3653-1308
James R Ault http://orcid.org/0000-0002-5131-438X
Frank Sobott https://orcid.org/0000-0001-9029-1865
Nigel J Savery http://orcid.org/0000-0002-0803-4075
Mark S Dillingham https://orcid.org/0000-0002-4612-7141

### Decision letter and Author response

Decision letter https://doi.org/10.7554/eLife.68829.sa1
Author response https://doi.org/10.7554/eLife.68829.sa2

## Additional files

### Supplementary files

• Transparent reporting form

### Data availability

All data generated or analysed during this study are included in the manuscript and supporting files. The HDX-MS mass spectrometry data have been deposited to the ProteomeXchange Consortium via the PRIDE partner repository with the dataset identifier PXD025332.

The following dataset was generated:

| Author(s) | Year | Dataset title | Dataset URL | Database and Identifier |
|---|---|---|---|---|
| Urrutia-Irazabal I, Ault JR, Sobott F, Savery NJ, Dillingham MS | 2021 | Analysis of the PcrA-RNA polymerase complex reveals a helicase interaction motif and a role for PcrA/UvrD helicase in the suppression of R-loops | https://www.ebi.ac.uk/pride/archive/projects/PXD025332/ | PRIDE, PXD025332 |

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
