## [Decision Letter]

**Acceptance summary:**

The resolution of R-loops that form during collisions between replication and transcription machineries is crucial for cell survival. This is exemplified by the lethality of deletion of PcrA, a helicase that appears to be involved in the resolution of such collisions. Here, the authors characterized the critical regions of PcrA/RNAP interactions and determined the function of such interactions. The manuscript's structural work is refined, elegant and leaves little room for doubt concerning the importance of the CTD PcrA-RNAP molecular interactions. This work moves the field forward in a meaningful way and unravels key aspects of PcrA/UvrD function with regards to interaction and function on RNAP. It will be of interest across the wide field of protein-DNA interactions, both prokaryotic and eukaryotic.

**Decision letter after peer review:**

Thank you for submitting your article "Analysis of the PcrA-RNA polymerase complex reveals a helicase interaction motif and a role for PcrA/UvrD helicase in the suppression of R-loops" for consideration by *eLife*. Your article has been reviewed by 3 peer reviewers, and the evaluation has been overseen by Maria Spies as a Reviewing Editor and Cynthia Wolberger as the Senior Editor. The reviewers have opted to remain anonymous.

Essential revisions:

1) All reviewers found the paper technically very strong and conceptually interesting, but thought that the clearance of R-loops by PcrA was not adequately, directly demonstrated – instead the evidence is circumstantial. I.e. seeing an increase in R-loop formation with the mutant PcrA does not prove PcrA clears R-loops. The reviewers though that it will be fine to accept this as a new hypothesis that warrants future testing. This needs to be pointed out in discussion. Another point for discussion is the potential role of NTD, as the CTD interaction is clearly not the only one.

2) The reviewers would also like to bring to the authors' attention a very recently published study https://dx.doi.org/10.3390%2Fcells10040935

While this work appeared while the authors manuscript was under review, it would be good to add to discussion.

*Reviewer #1 (Recommendations for the authors):*

This is one of the better papers I have reviewed all year and I have no corrections to report! Wonderfully executed and written up!

*Reviewer #2 (Recommendations for the authors):*

The manuscript's structural work is refined, elegant and leaves little room for doubt concerning the importance of the CTD PcrA-RNAP molecular interactions. However, I feel that the biological connotations of this newly characterized interaction are a bit premature, with the proposed models relying heavily on implications derived from their structural data. Nevertheless, this work moves the field forward and will be impactful, especially if some of the following suggestions are addressed.

The first half of the paper characterizing the PcrA-RNAP complex is very well-designed and controlled, and the arguments for PcrA Tudor-domain-mediated binding to rpoB SI1 are convincing. Direct binding of PcrA to TECs shown through EMSAs and chromatography leaves little doubt that the CTD domain is crucial for the PcrA-RNAP interaction. A revision we would recommend for this part of the study is running a similar set of experiments using the 2A domain of PcrA as a binding partner, as the authors show later in the paper that this domain interacts with RNAP seemingly independently of the CTD. The identification of SI1 and Tudor domains as the binding sites through HDX was well-conceived and unbiased – I am convinced that these are the only regions of significant interaction between PcrA-CTD and RNAP. The characterization of the specific interaction motif of the Tudor domain and the identification of analogous motifs in known binding partners of PcrA, specifically UvrD, present compelling evidence of the importance of the apical TGE for protein-protein interaction. HDX data indicating full-length PcrA interaction near the sites of the RNA and DNA exit channels was intriguing, but I would have liked to see additional validation of 2A domain interaction through EMSAs or chromatography, as shown for CTD data.

While the structural data was resounding, I was less convinced regarding the implication and impact of the biological data presented in the paper. As PcrA was previously known to translocate 3' to 5' along ssDNA and unwind dsDNA, the experiments demonstrating the necessity of a 3' DNA overhang present a fundamental shift in our understanding of helicase function. While the demonstration of PcrA unwinding DNA:RNA hybrids is novel, based on previous data demonstrating that PcrA/UvrD is capable of overcoming transcription barriers, this is not necessarily unexpected.

Further experiments into the role of PcrA as a resolution factor in R-Loops left me with some unclear conclusions. While in vitro experiments demonstrate that free PcrA is sufficient for unwinding DNA:RNA hybrids, in vivo work appears to suggest that interaction of the CTD with RNAP is crucial for this activity. The authors don't address this dichotomy, nor was there a discussion on how to reconcile this information with previous reports of the minimal defects reported in CTD-deleted strains. The authors cite PcrA-NTD's theoretical ability to bind to RNAP and function independently of CTD as a possible explanation for viability of ΔCTD mutants, but do not show either structural, nor biological information in support of this.

Lastly, neither model proposed by the authors is reconciled by their assertion that the helicase core interacts closely with the RNA and DNA exit channels of RNAP – both models require PcrA to lag significantly behind RNAP, allowing for R-Loops to form on the template strand in between PcrA and RNAP. However, the authors' own HDX data demonstrates that the helicase core binds near these channels, leading me to question if there is sufficient physical space for an R-Loop to form. The authors also cite computational data demonstrating the PcrA linker length is generally proportional to the distance between the SI1 and the exit channels – postulating that the exit channels interaction is present across species, and a vital aspect of PcrA-RNAP interaction.

Lastly, while the structural component of the manuscript demonstrating the PcrA-CTD interaction with RpoB is well-rounded, a lack of information on the NTD of PcrA, as well as contradictory biological results on the capacity of PcrA to process DNA:RNA hybrids independently of RNAP leave me with a less than clear understanding of the models proposed by the authors. I would recommend performing some of the experiments done with the CTD with the NTD as well in order to provide a complete picture of PcrA-RNAP interaction. If both previously identified domains are included in this manuscript, it would be a great step forward in the field.

*Reviewer #3 (Recommendations for the authors):*

1) Cytiva is now GE Healthcare.

2) Authors need to explain the C-protease is His-tagged to make the methods clearer.

3) For HDX methods: What were the number of technical and biological repeats performed (Masson 2017)? The 5.5 fold dilution into the deuterated buffer would reduce the PcrA to below the Kd. Why was this chosen, since the protein would not be binding as well and therefore the protection limited?

4) Figure 1A: can the authors explain the multiple high MW bands in the presence of PcrA and TEC?

5) Does PcrA-CTD bind to a DNA/RNA hybrid? This might help the readers understand if PcrA needs to be in close association with RNAP to function, since it is not a lethal construct.

6) Figure S1D shows bandshifts but the shifts are not easy to see. Showing the bandshift induced by CTD in the same gel would help. The argument that the fluorescence of the labeled PcrA is lower is difficult to be sure of. How many times was this repeated and is the intensity quantified?

7) It is really interesting how the PcrA CTD binding motif is found across species, but in *B.Subtilis* it is in a lineage specific insertion domain. Do all species with this sequence have PcrA, a cross correlation would be useful. How does UvrD fit in with this?

8) Figure 3D, Mfd does not diminish binding of RNAP: Doesn't Mfd need to be de-repressed to bind RpoB? Is there any evidence that they are binding to different locations? This is no clear evidence in this assay. There is no indication that Mfd is binding RpoB. Wouldn't it be better to titrate in the RID of Mfd? The evidence for Mfd and PcrA binding different locations on RNAP is not apparent here.

9) Figure 6C would be well supported by an ATPase assay of a RNA/DNA versus dsDNA substrate. Using an ssDNA substrate does not show whether the removal of RNA is efficient.

10) Section starting "Induction of PcrA-E224Q from an ectopic locus…": Please use PcrA-E224Q throughout when referring to the mutant, the use of PcrA alone was confusing.

11) Did the authors perform the HDX experiments on PcrA-ΔCTD without the E224Q mutant? This would address whether such mutants still interact with RNAP and clarify the mechanism a little more.

12) Does PcrA-ΔCTD increase R-loop formation?

13) Is there any significance to the elevated survival of mfd+PcrA E224Q relative to PcrA E224Q of Figure S10A?

14) Figure S12: How can the authors be sure that the protection seen with full length PcrA is not due to a conformational change in the RNAP? Rather than a long linker putting PcrA onto the DNA/RNA pockets of RNAP? Isn't it also surprising that the linker doesn't show up on HDX? Isn't there a substantial entropic cost to the formation of such a long loop?

15) The authors need to acknowledge and cite the work from a highly-pertinent recent publication that they may have missed: https://dx.doi.org/10.3390%2Fcells10040935

---

## [Author Response]

Essential revisions:1) All reviewers found the paper technically very strong and conceptually interesting, but thought that the clearance of R-loops by PcrA was not adequately, directly demonstrated – instead the evidence is circumstantial. I.e. seeing an increase in R-loop formation with the mutant PcrA does not prove PcrA clears R-loops. The reviewers though that it will be fine to accept this as a new hypothesis that warrants future testing. This needs to be pointed out in discussion.

We agree and have altered than manuscript accordingly. Although our work provides extensive new information on the PcrA:RNAP interface, and makes a clear link between PcrA activity and the steady state level of R-loops in cells, it does not show directly that PcrA unwinds co-transcriptional R-loops. That is our hypothetical model based upon three key observations; (1) the core motor domains of PcrA are close to the RNA/DNA exit channels, (2) PcrA is capable of unwinding DNA:RNA hybrids in vitro and, (3) interfering with PcrA activity in vivo leads to R-loop accumulation. We have re-written parts of the introduction (pg 4, line 31) and discussion (pg 16, line 30; pg18, line 3; pg 18, line 29) to more clearly acknowledge this, and now describe our final cartoons of the mechanism as “hypothetical models” (Figure 7 legend). In future experiments we aim to test these models directly by biochemical reconstitution of co-transcriptional R-loop formation and subsequent resolution by PcrA.

Another point for discussion is the potential role of NTD, as the CTD interaction is clearly not the only one.

We did not perform any experiments with the helicase core (lacking the CTD) because our published experiments (Gwynn et al., 2013; Sanders et al., 2017) show that it does not interact with RNAP in vitro and barely interacts with the TEC (this work; Kd >> 1.5 µM). We expand upon this point in our response to the individual referees below and have amended the results (pg 11, line 12) and discussion as requested (pg 17, line 19).

2) The reviewers would also like to bring to the authors' attention a very recently published study https://dx.doi.org/10.3390%2Fcells10040935While this work appeared while the authors manuscript was under review, it would be good to add to discussion.

We accept this point and have included a discussion of the relevant paper as requested (pg 19, line 10).

Reviewer #2 (Recommendations for the authors):The manuscript's structural work is refined, elegant and leaves little room for doubt concerning the importance of the CTD PcrA-RNAP molecular interactions. However, I feel that the biological connotations of this newly characterized interaction are a bit premature, with the proposed models relying heavily on implications derived from their structural data. Nevertheless, this work moves the field forward and will be impactful, especially if some of the following suggestions are addressed.The first half of the paper characterizing the PcrA-RNAP complex is very well-designed and controlled, and the arguments for PcrA Tudor-domain-mediated binding to rpoB SI1 are convincing. Direct binding of PcrA to TECs shown through EMSAs and chromatography leaves little doubt that the CTD domain is crucial for the PcrA-RNAP interaction. A revision we would recommend for this part of the study is running a similar set of experiments using the 2A domain of PcrA as a binding partner, as the authors show later in the paper that this domain interacts with RNAP seemingly independently of the CTD. The identification of SI1 and Tudor domains as the binding sites through HDX was well-conceived and unbiased – I am convinced that these are the only regions of significant interaction between PcrA-CTD and RNAP. The characterization of the specific interaction motif of the Tudor domain and the identification of analogous motifs in known binding partners of PcrA, specifically UvrD, present compelling evidence of the importance of the apical TGE for protein-protein interaction. HDX data indicating full-length PcrA interaction near the sites of the RNA and DNA exit channels was intriguing, but I would have liked to see additional validation of 2A domain interaction through EMSAs or chromatography, as shown for CTD data.

It is far from straightforward to make a 2A-only construct for PcrA because 2A contains the entire 2B domain as an insertion. Nevertheless, the general point being made here is similar to that of reviewer 3: that we should study the interactions with the helicase core independently of the CTD domain. We specifically did not perform this HDX experiment because we already knew from published experiments that PcrA lacking the CTD does not interact with RNA polymerase in vitro. Moreover, in this work, we show that even with the TEC, an interaction with helicase core is barely detectable (Figure 1A). Thus, it appears that the interaction of the helicase core with RNAP requires the presence of the CTD, either because it is too weak alone, or because the CTD-SI1 interaction facilitates the interaction allosterically. We have added to the results (pg 11, line 12) and discussion (pg 17, line 19) to better explain these points.

While the structural data was resounding, I was less convinced regarding the implication and impact of the biological data presented in the paper. As PcrA was previously known to translocate 3' to 5' along ssDNA and unwind dsDNA, the experiments demonstrating the necessity of a 3' DNA overhang present a fundamental shift in our understanding of helicase function. While the demonstration of PcrA unwinding DNA:RNA hybrids is novel, based on previous data demonstrating that PcrA/UvrD is capable of overcoming transcription barriers, this is not necessarily unexpected.

Our in vitro helicase activities were only designed to test whether, in principle, PcrA could unwind RNA secondary structure and/or R-loops by asking if it was able to unwind short RNA duplexes or DNA:RNA hybrids respectively. We showed that PcrA cannot translocate along RNA, nor can it unwind RNA duplexes and we can therefore dispense with the idea that it might act as an RNA chaperone. We also showed that PcrA can unwind DNA:RNA hybrids, which is consistent with (but we agree does not prove) R-loop unwinding activity in vivo. Moreover, the lack of translocation activity on ssRNA also limits the models for R-loop unwinding that are feasible (eg R-loop unwinding could not proceed by 3’-5’ translocation on nascent RNA, but would have to result from ssDNA translocation on either the transcribed or non-transcribed strands as per our cartoon; Figure 7). Therefore, while we agree that these results may not be that surprising *per se*, we think they are valuable in providing support for the model we develop for PcrA as an R-loop surveillance factor. We also agree that previous landmark papers (showing that UvrD and PcrA are important for overcoming transcription impediments to replication) might indeed have led one to speculate that PcrA can unwind R-loops. Our paper provides experimental evidence for this idea, which is just one way among many in which a SF1 helicase might conceivably suppress replication:transcription conflicts.

Further experiments into the role of PcrA as a resolution factor in R-Loops left me with some unclear conclusions. While in vitro experiments demonstrate that free PcrA is sufficient for unwinding DNA:RNA hybrids, in vivo work appears to suggest that interaction of the CTD with RNAP is crucial for this activity. The authors don't address this dichotomy, nor was there a discussion on how to reconcile this information with previous reports of the minimal defects reported in CTD-deleted strains. The authors cite PcrA-NTD's theoretical ability to bind to RNAP and function independently of CTD as a possible explanation for viability of ΔCTD mutants, but do not show either structural, nor biological information in support of this.

We do not agree that a dichotomy exists between the ability of PcrA alone to unwind model DNA:RNA hybrids, and the apparent importance of the CTD-RNAP interaction for modulating R-loops in vivo. In previous work (Gwynn et al., 2013) we have shown that the helicase activity of PcrA is not substantially affected by either the removal of the CTD or the presence of RNAP. Therefore, we do not imagine that formation of the CTD-RNAP interface catalytically *activates* the core helicase for R-loop unwinding, but rather that it helps to *target* the helicase to its physiological substrate in vivo. In this view, the role of the CTD is simply to increase the local concentration of PcrA near R-loops. We have clarified this in the discussion (pg 18, line 17). To test our idea, we plan to biochemically reconstitute the formation of co-transcriptional R-loops with *Bacillus subtilis* RNAP and then test the ability of PcrA with and without the CTD to resolve them. These experiments are quite complex and beyond the scope of the current paper.

The reviewer is correct to bring up the (lack of) phenotypes of the PcrA-ΔCTD mutant. We argue in the paper that this might be explained by residual interaction between the helicase core and RNAP that supports R-loop resolution, and we do provide experimental support for this assertion in the form of the HDX experiments with full length PcrA. However, we acknowledge that this argument is not entirely satisfying because, as discussed above, we know from in vitro work that this interface must be very weak. We were only able to probe the nature of this interaction by HDX in the context of full length PcrA, where it is presumably stabilised by the CTD-SI1 interaction. This complexity, in concert with the essential nature of the *pcra* gene, underlies our approach of using “dominant negative” interventions (i.e. helicase-dead PcrA or free CTD) in order to provoke phenotypes in vivo. We suspect that by actively blocking (rather than simply removing) the activity of PcrA, we are stabilising R-loops that would otherwise have been unwound not only by wild type PcrA, but also by other degenerate mechanisms for R-loop resolution. We have added some more discussion to the Results section (pg 17, line 19) to better explain these points.

Lastly, neither model proposed by the authors is reconciled by their assertion that the helicase core interacts closely with the RNA and DNA exit channels of RNAP – both models require PcrA to lag significantly behind RNAP, allowing for R-Loops to form on the template strand in between PcrA and RNAP. However, the authors' own HDX data demonstrates that the helicase core binds near these channels, leading me to question if there is sufficient physical space for an R-Loop to form. The authors also cite computational data demonstrating the PcrA linker length is generally proportional to the distance between the SI1 and the exit channels – postulating that the exit channels interaction is present across species, and a vital aspect of PcrA-RNAP interaction.

We thank the reviewer for this comment, which arises from the way we have drawn the cartoons of our proposed mechanisms for R-loop unwinding. These were intended to convey, in a simple manner, the idea that the R-loop could potentially be unwound in two ways, direct RNA:DNA unwinding in the same direction as transcription or by backtracking the RNAP to indirectly separate the hybrids. In the first model, the PcrA may act immediately behind the RNAP, to resolve R-loops *as they are formed*, and this would be consistent with close contact between the 2A domain and the exit channels as the reviewer correctly points out. Nevertheless, given that the helicase core binds weakly to the RNAP and is attached via an extended linker to the CTD-SI1 anchor point, there may also be some scope for the helicase to detach from the RNAP to unwind longer hybrids. The complete lack of structural insight into what R-loops actually look like in practice makes it difficult to be anything more than quite speculative at this stage! In the second model, the helicase would need to face “away” from the RNAP which, as we explain in the manuscript, is inconsistent with the protection of the 2A domain which is at the leading edge of the motor protein. We do not favour this model, but wanted to include it as a formal possibility because other groups (including our own) have provided evidence that PcrA/UvrD can backtrack the TEC. We have added a short section to the discussion and have modified the relevant figure legend to explain our position (pg 18, line 17; Figure 7 legend).

Lastly, while the structural component of the manuscript demonstrating the PcrA-CTD interaction with RpoB is well-rounded, a lack of information on the NTD of PcrA, as well as contradictory biological results on the capacity of PcrA to process DNA:RNA hybrids independently of RNAP leave me with a less than clear understanding of the models proposed by the authors. I would recommend performing some of the experiments done with the CTD with the NTD as well in order to provide a complete picture of PcrA-RNAP interaction. If both previously identified domains are included in this manuscript, it would be a great step forward in the field.

This comment repeats two earlier points and we refer to our answers above. Briefly, there is no contradiction between the biological results and the helicase assays. We envisage the CTD-SI1 as an anchoring interaction that targets PcrA to its physiological substrate, as opposed to a mechanism for allosteric control of unwinding activity. Experiments were not performed with the PcrA-ΔCTD construct in vitro as it does not bind to RNAP at tractable concentrations.

Reviewer #3 (Recommendations for the authors):1) Cytiva is now GE Healthcare.

This has been corrected throughout.

2) Authors need to explain the C-protease is His-tagged to make the methods clearer.

The requested change has been made.

3) For HDX methods: What were the number of technical and biological repeats performed (Masson 2017)? The 5.5 fold dilution into the deuterated buffer would reduce the PcrA to below the Kd. Why was this chosen, since the protein would not be binding as well and therefore the protection limited?

Four technical repeats were performed, each using the same protein sample that was pooled from different batches of purified proteins that had been buffer exchanged into the HDX-MS buffer before transferring the samples to the robot. After the dilution, PcrA is approximately 400 nM which is around the Kd value based on EMSA and above the K_d_ value we have obtained previously from SPR measurements (Gwynn et al., 2013). This value was chosen because, in the absence of DNA, PcrA tends to precipitate at higher concentrations in the buffer used for HDX. Moreover, pilot experiments indicated that higher protein concentrations saturated the detector. We have clarified the number of technical repeats and also included a reference to the PRIDE database accession number for the raw data in the Methods section (pg 22, line 29).

4) Figure 1A: can the authors explain the multiple high MW bands in the presence of PcrA and TEC?

We don’t have a clear explanation for the presence of additional low occupancy bands in the EMSAs, which are most noticeable for the CTD-dependent shift. We don’t think this reflects multiple PcrA binding events because the HDX data shows such a single and small region of protection. It might reflect PcrA binding/stabilising different RNAP-TEC conformations.

5) Does PcrA-CTD bind to a DNA/RNA hybrid? This might help the readers understand if PcrA needs to be in close association with RNAP to function, since it is not a lethal construct.

The PcrA-CTD does not bind to DNA (Sanders et al., 2017). However, based on the second sentence, we suspect the reviewer means to ask “does PcrA-ΔCTD bind to a hybrid”. We do not know, but expect it will both bind and unwind hybrids, because this enzyme has near wild type ATPase and helicase activity on DNA-only substrates (Velankar et al., 1999; Gwynn et al., 2013). This is related to the question of whether hybrid unwinding requires engagement of the CTD with the SI1 domain, and to our discussion (Reviewer 2, point 3) in which we distinguish between the concept of the CTD either targeting or activating the helicase. We have clarified this point in the discussion (pg17, line 19 and pg18, line 17).

6) Figure S1D shows bandshifts but the shifts are not easy to see. Showing the bandshift induced by CTD in the same gel would help. The argument that the fluorescence of the labeled PcrA is lower is difficult to be sure of. How many times was this repeated and is the intensity quantified?

We agree that the change in bandshift position is difficult to evaluate in the absence of the CTD-only bandshift. In fact, this gel did include a CTD-only shift which was used in Figure 1A. We now show the whole gel in Figure 1 – Supp 1D which helps one to appreciate that the labelled TEC changes position to the lower mobility associated with the CTD supershift. We also add the quantification of the amount of full length PcrA in the shifted band which is substantially reduced in the presence of free CTD. This experiment with the labelled PcrA was not repeated but supports the conclusion that full length PcrA and CTD compete for binding to RNAP, a point which is perhaps more convincingly made by data presented in the main paper using a different approach (Figure 6F).

7) It is really interesting how the PcrA CTD binding motif is found across species, but in *B.Subtilis* it is in a lineage specific insertion domain. Do all species with this sequence have PcrA, a cross correlation would be useful. How does UvrD fit in with this?

To the best of our knowledge, the PcrA/UvrD helicase is almost ubiquitous in bacterial species (Gilhooly et al., 2013). Therefore, the answer is yes; all bacteria with the interaction motif in RNAP also have a PcrA orthologue. In *E. coli*, the positioning of the motif is slightly different but *effectively* within the same domain (Figure 2 – Supp 2) although that domain has been named differently, is variable between species, and is therefore considered “lineage-specific”. The implication of our work is that the function of this domain is in fact the same across species, and that the function is to bind PcrA/UvrD.

8) Figure 3D, Mfd does not diminish binding of RNAP: Doesn't Mfd need to be de-repressed to bind RpoB? Is there any evidence that they are binding to different locations? This is no clear evidence in this assay. There is no indication that Mfd is binding RpoB. Wouldn't it be better to titrate in the RID of Mfd? The evidence for Mfd and PcrA binding different locations on RNAP is not apparent here.

This is a fair point. Our experiment is based on an underlying assumption that the free Mfd we titrate into the pulldown as a competitor is able to bind RNAP from the extract, and (originally) we showed no evidence to support this assumption. Although it is certainly expected that full length Mfd would interact with a TEC, it is perhaps not immediately clear that Mfd would bind to RNAP in the nucleic-acid depleted extracts used here. We now include (New Table 2, and new methods section) the results of an Mfd proteomics experiment in which recombinant biotinylated Mfd (as bait) pulls down *Bacillus* RNAP subunits from a cell extract under similar conditions. This justifies the assumption that the Mfd we added (Figure 3) was able to interact with RNAP and is therefore consistent with the idea that PcrA and Mfd bind to different sites on RNAP. Importantly however, the conclusion that Mfd and PcrA bind at different locations also arises from other results in our paper. The location for the Mfd-RNAP interaction is well-characterised structurally. Our HDX experiments show that PcrA binds to different locations on the surface of RNAP but also that there is no protection of the β1 domain to which Mfd binds based on the *E. coli* model. We now make these points in the Results section (page 10, line 25).

9) Figure 6C would be well supported by an ATPase assay of a RNA/DNA versus dsDNA substrate. Using an ssDNA substrate does not show whether the removal of RNA is efficient.

We don’t understand this point. An ATPase assay would not report on RNA removal. Instead, our helicase (strand displacement) assays show that PcrA efficiently unwinds a DNA:RNA hybrid. The ATPase assays were used as a proxy to determine whether the protein can translocate on RNA (or only on ssDNA as expected) and are informed by our current understanding of the inchworm mechanism employed by SF1 helicases. Together the data strongly suggest that the model hybrid substrate must be unwound as a result of translocation on the DNA strand coupled to strand displacement of the complementary RNA. This observed behaviour is also consistent with the well-known requirement for a 3’-overhang on the translocating strand to facilitate unwinding.

10) Section starting "Induction of PcrA-E224Q from an ectopic locus…": Please use PcrA-E224Q throughout when referring to the mutant, the use of PcrA alone was confusing.

We thank the reviewer for pointing this out and have corrected the text accordingly.

11) Did the authors perform the HDX experiments on PcrA-ΔCTD without the E224Q mutant? This would address whether such mutants still interact with RNAP and clarify the mechanism a little more.

As explained above, we know that PcrA-CTD does not bind to RNAP in vitro with sufficient avidity to facilitate HDX experiments, so we did not perform them.

12) Does PcrA-ΔCTD increase R-loop formation?

We have not performed this experiment. As explained above, based on the lack of a clear interaction between the PcrA-ΔCTD and RNAP (that we argue might explain the lack of phenotypes associated with the mutation), we opted instead to use a “dominant negative interference” approach to elicit phenotypes rather than mutating or deleting endogenous PcrA.

13) Is there any significance to the elevated survival of mfd+PcrA E224Q relative to PcrA E224Q of Figure S10A?

The difference here is very small indeed in comparison with the dramatic differences in cell viability caused by the PcrA mutation. The data hints that removing Mfd alleviates to some extent the toxic effects of dysfunctional PcrA, but we think it would be unwise to try and draw any firm conclusions about interactions between *mfd* and *pcra* based on this data.

14) Figure S12: How can the authors be sure that the protection seen with full length PcrA is not due to a conformational change in the RNAP? Rather than a long linker putting PcrA onto the DNA/RNA pockets of RNAP? Isn't it also surprising that the linker doesn't show up on HDX? Isn't there a substantial entropic cost to the formation of such a long loop?

The reviewer makes a fair point. Fundamentally, HDX-MS can report both protein interactions and conformational changes through changes in rates of H/D exchange upon protein:protein interactions. However, we think our interpretation is reasonable based on the following logic. We have HDX-data for both the CTD and full length PcrA. We know the small CTD binds tightly to RNAP alone, and the HDX shows a small region of protection in the SI1 domain which we confirm is the interaction site by mutagenesis. The HDX for full length protein shows the same interaction site at SI1 plus additional regions of protection around the DNA/RNA exit channels. These regions of HDX protection could, in principle, be caused either by allosterically-induced conformational changes, and/or by simple protection due to new protein:protein interfaces (the latter being our interpretation). For the first scenario to be exclusively true, one would have to argue that the N-terminal core region of the helicase causes conformational changes at the exit channels by binding elsewhere on the protein surface while simultaneously leaving no trace for its true physical interaction. Any putative conformational change cannot have been caused at a distance by binding of the CTD because we would then expect to see the same signals in the CTD-only data. We cannot exclude the idea that the helicase core binds near the exit channels and also causes conformational changes in this region which are partially responsible for the signals we observe. As discussed in the manuscript, our interpretation is also corroborated by crosslinking data for UvrD-RNAP from the Nudler laboratory.

Regarding the lack of a protection signal for the linker region. This reflects the fact that the linker peptides were not detected in the analysis above a quality threshold (Figure 4A; already discussed in original manuscript). We agree that there would be an entropic cost associated with the restriction of the linker into the extended position we postulate, but presumably this could be offset by binding energy. In the absence of a high-resolution structure of the PcrA-RNAP complex (which we are pursuing), the conformation adopted by the linker is still a matter of speculation. Our intention with the crude modelling shown in Figure 7 – Supp 1 was only to show that the linker was long enough to bridge the two interaction sites on RNAP we had detected by HDX-MS.

15) The authors need to acknowledge and cite the work from a highly-pertinent recent publication that they may have missed: https://dx.doi.org/10.3390%2Fcells10040935

This is point 2 in the “Essential Revisions” section. This paper had not been published when we submitted our manuscript. We agree that it should be cited and appropriately discussed, and have modified the text accordingly (page 19, line 10).

References:

Gilhooly NS, Gwynn EJ, Dillingham MS. Superfamily 1 helicases. Front Biosci (Schol Ed). 2013 Jan 1;5:206-16. doi: 10.2741/s367. PMID: 23277046.

Gwynn EJ, Smith AJ, Guy CP, Savery NJ, McGlynn P, Dillingham MS. The conserved C-terminus of the PcrA/UvrD helicase interacts directly with RNA polymerase. PLoS One. 2013 Oct 16;8(10):e78141. doi: 10.1371/journal.pone.0078141. PMID: 24147116; PMCID: PMC3797733.

Sanders K, Lin CL, Smith AJ, Cronin N, Fisher G, Eftychidis V, McGlynn P, Savery NJ, Wigley DB, Dillingham MS. The structure and function of an RNA polymerase interaction domain in the PcrA/UvrD helicase. Nucleic Acids Res. 2017 Apr 20;45(7):3875-3887. doi: 10.1093/nar/gkx074. PMID: 28160601; PMCID: PMC5397179.

Velankar SS, Soultanas P, Dillingham MS, Subramanya HS, Wigley DB. Crystal structures of complexes of PcrA DNA helicase with a DNA substrate indicate an inchworm mechanism. Cell. 1999 Apr 2;97(1):75-84. doi: 10.1016/s0092-8674(00)80716-3. PMID: 10199404.